

# Developing a representative snow monitoring network in a forested mountain watershed

Kelly E. Gleason[1], Anne W. Nolin[1], Travis R. Roth[1]

[1] College of Earth, Ocean, and Atmospheric Sciences, Oregon State University, Corvallis, Oregon, 97331, USA

*Correspondence to*: Kelly E. Gleason (kgleason@usgs.com)

**Abstract.** Current snow monitoring networks may not be representative of basin-scale distributions of snow water equivalent (SWE), especially in areas where forests and snowpacks are changing. A challenge in establishing new ground-based stations for monitoring snowpack accumulation and ablation is to locate the sites in areas that represent the key processes affecting snow accumulation and ablation. This is especially challenging in forested montane watersheds where the combined effects of terrain, climate, and land cover affect seasonal snowpack. The objectives of this research were to identify the key physiographic drivers of SWE, classify the landscape based on those physiographic drivers, and use that classification to identify a parsimonious set of monitoring sites in a forested watershed in the western Oregon Cascades mountain range. We used a binary regression tree (BRT) non-parametric statistical model to classify 1 April SWE. Training data for the BRT classification were derived using spatially distributed estimates of SWE from a validated physically-based model of snow evolution. The optimal BRT model showed that elevation, vegetation type, and vegetation density were the most significant drivers of SWE in the watershed. Geospatial elevation and land cover data were used to map the BRT-derived snow classes across the watershed. Specific snow monitoring sites were selected randomly within the BRT-derived snow classes to capture the range of spatial variability in snowpack conditions in the McKenzie River Basin. The Forest Elevational Snow Transect (ForEST) represents combinations of forested and open land cover types at low, mid, and high elevations. After five years of snowpack monitoring, the ForEST network provides a valuable and detailed dataset of snow accumulation, snow ablation, and snowpack energy balance in forested and open sites from the rain-snow transition zone to upper seasonal snow zone in the western Oregon Cascades.



# 1 Introduction

Mountain snowpack is declining as a result of the warming climate (Kunkel et al., 2016; Knowles, 2015; Pederson et al., 2013; Rupp et al., 2013; Pederson et al., 2011; Mote, 2006), subsequently shifting timing (Fritze et al., 2011; Clow, 2010) and volume of streamflow (Woodhouse et al., 2016; Berghuijs et al., 2014; Luce and Holden, 2009) across the western United States. Luce et al., (2013) argue that the declining snowpack is also the result of weakening westerlies leading to a decline in mountain precipitation in the interior West. The volume and seasonality of water produced from these snow-dominated watersheds varies spatially and temporally as a function of precipitation and temperature (Tennant et al., 2015; Barnett et al., 2005; Regonda et al., 2005), as well as local physiographic effects of topography, geology, and vegetation dynamics (Molotch and Meromy, 2014; Clark et al., 2011; Jefferson et al., 2008; Ffolliott et al., 1989).

Montane snow-dominated river basins are topographically complex. Elevation, slope, aspect, and exposure influence snowpack dynamics across a watershed through alterations of precipitation amount and phase (rain vs. snow), wind speed, temperature, and humidity. The degrees to which these physiographic variables control snow persistence vary as functions of snow accumulation and snow ablation, from the plot to regional spatial scales (López-Moreno et al., 2015; Biederman et al., 2014; López-Moreno et al., 2013; Deems et al., 2006; Molotch and Bales, 2005), and from daily to seasonal scales (Fassnacht et al., 2012; Jepsen et al., 2012). In the Pacific Northwest, montane basins are a successional patchwork of variable forest cover driven by forest harvest and replanting, pest infestations, and fire disturbance. In forested regions, snow accumulation and ablation processes are strongly influenced by vegetation structure (Veatch et al., 2009; Musselman et al., 2008; Jost et al., 2007; Trujillo et al., 2007; Sicart et al., 2004; Murray and Buttle, 2003; Pomeroy et al., 2002; Link and Marks, 1999). Both vegetation and topography influence the distribution of solar radiation (Musselman et al., 2015; Musselman et al., 2012; Davis et al., 1997; Dozier, 1980;), snow-surface albedo (Gleason and Nolin, 2016; Gleason et al., 2013; Molotch et al., 2004; Melloh et al., 2002), net longwave radiation (Lundquist et al., 2013; Sicart et al., 2004) , wind speed (Winstral and Marks, 2002) and turbulent fluxes (Burns et al., 2014; Garvelmann et al., 2014; Marks et al., 2008).

Snow water equivalent (SWE) is a critical hydrologic resource in the montane western US that has been actively monitored for decades by the Natural Resources Conservation Service (NRCS). The NRCS manages approximately 858 Snowpack Telemetry (SNOTEL) stations across the western US



(http://www.wcc.nrcs.usda.gov/snotel/SNOTEL_brochure.pdf). These stations provide near real-time measurements of SWE, temperature, and precipitation; essential data for operational streamflow forecasts used by water managers who balance a wide range of needs including irrigation, aquatic habitat, hydropower, recreation, and municipal water use. While most SNOTEL sites have been operating since the early 1980s, the data are meant to be used as indices to forecast discharge. These records are valuable but the stations were not designed to be nor are they representative of the total snow in a basin (Meromy et al., 2013; Molotch and Bales, 2006a). Also, they may not be representative of snow conditions under future climate. In the Oregon Cascades, the SNOTEL monitoring network stations are located within a narrow elevation range (1140–1510 m) that may not capture the inherent variability in the spatial distribution of snow under present day or warmer climate conditions (Nolin, 2012; Brown, 2009).

In the rugged, forested, and frequently cloud-covered montane watersheds of the Pacific Northwest, modeling has been shown to be an effective means of augmenting remote sensing, and a valuable tool for predicting snow conditions under climate change (Sproles et al., 2013; Tague and Grant, 2009; Veatch et al., 2009; Luce et al., 1999; Cline et al., 1998). Landscape characteristics have been used to predict snowpack conditions at hillslope scales using non-parametric binary regression tree (BRT) statistical classification models (Molotch et al., 2005; Anderton et al., 2004; Erxleben et al., 2002; Winstral et al., 2002; Balk and Elder, 2000; Elder et al., 1998). Larger scale BRT approaches have also been conducted using remotely sensed snow-covered area and interpolation methods (Molotch and Meromy, 2014; Molotch and Bales, 2006b). However, no study to date has used landscape characteristics in conjunction with modeled and validated physically-based and spatially distributed SWE data to understand physiographic drivers of snow accumulation at broad scales (watersheds > 1000 km$^2$) and to identify optimal locations for snowpack monitoring. Additionally, most of the research on the physiographic relationships to snow processes has been done in cold-dry continental snowpacks where mid-winter melt events are infrequent and wind redistribution is substantial (Molotch et al., 2005; Erxleben et al., 2002; Winstral et al., 2002; Balk and Elder, 2000). Much less is known about how physiographic conditions influence the temperature sensitive snowpacks in the forested maritime basins of the Pacific Northwest. This paper evaluates the existing snow monitoring network in the McKenzie River Basin within the context of a projected future warming climate, and presents an objective methodology for site selection of a snow monitoring network that captures the spatial variability in snow accumulation in a montane forested watershed in the western Oregon Cascades.

In order to develop a representative snow monitoring network, the objectives of this research were the following:



1. Determine the key physiographic drivers of spatial variability in snow accumulation;

2. Classify snow classes in the watershed based on key physiographic drivers using a non-parametric statistical model;

3. Spatially distribute these snow classes across the watershed using a geospatial model;

4. Select site locations for a snow monitoring network which spans the spatial variability in snow water equivalent in the McKenzie River Basin.

## 2 Methods

### 2.1 Description of the Study Site

In the heart of the western Oregon Cascades, the McKenzie River is a major tributary of the Willamette River (Figure 1). The McKenzie River Basin (MRB) drains an area of 3,041 km$^2$, and covers about 12% of the land area in the greater Willamette River Basin. The MRB is a densely forested mountainous watershed, ranging in elevation from 150 m to 3150 m, that is a managed for timber production throughout much of the seasonal snow zone. Brooks et al., (2012), determined that 60-80 % of summer flow in the Willamette River originated from elevations above 1200 m in the Oregon Cascades. The porous young volcanic geology in these mountains allows much of the snowmelt to percolate into groundwater systems (Tague and Grant, 2009; Jefferson et al., 2008; Tague and Grant, 2004). The snowmelt-fed groundwater-supplied McKenzie River provides 25 % of the late season volumetric base flow to the Willamette River at its confluence with the Columbia River (Hulse et al., 2002).

### 2.2 Description of the Data

Gridded data were obtained for physiographic variables shown in the literature to influence snow accumulation and ablation, including elevation, slope, aspect, incoming solar radiation, wind, and three vegetation variables from the following sources for the extent of the MRB. A Digital Elevation Model (DEM) was obtained from the National Elevation Dataset at a 10-m resolution. Slope, aspect, and incoming solar radiation were calculated from the DEM using the Spatial Analyst and Solar Radiation toolboxes in ArcGIS 10.1 (ESRI, Redlands, CA). Upwind contributing area data, which captures the variability in snow deposition as a result of wind redistribution for each cell throughout the watershed (Winstral et al., 2002), was calculated following Molotch et al., (2005). The 2006 National Land Cover Database (NLCD) was used to classify land



cover across the watershed (Fry et al., 2011). The US Geological Survey (USGS) LANDFIRE Data Distribution Site provided the Existing Vegetation – Percent Canopy Cover (EVC) data at 30-m spatial resolution. Normalized Difference Vegetation Index (NDVI) data were obtained from the Moderate Resolution Imaging SpectroRadiometer (MODIS) MOD13Q1 – Vegetation Indices, 16-day Land Product for the earliest date possible

in April 2009, at a 250-m spatial resolution. Watershed boundaries were defined using the USGS National Hydrography Dataset. Public land ownership data were provided by the Oregon Department of Forestry, and obtained from the website, http://www.oregon.gov/odf/pages/gis/gisdata.aspx.

        Modeled and gridded SWE data across the MRB (Figure 2) were provided by (Sproles et al., 2013). These data were developed using a physically-based spatially distributed snow mass and energy balance model,

SnowModel (Liston and Elder, 2006). SnowModel uses micrometeorological and topographic data to distribute snow across the landscape accounting for climatic, topographic, and vegetation variability. The model was modified by Sproles et al., (2013) to account for rain/snow precipitation phase partitioning, and snow albedo decay in forested landscapes. This model was calibrated and validated using data from the four SNOTEL sites, meteorological data from the HJ Andrews Long Term Ecological Research site and National Weather Service

stations and Landsat fractional snow covered area data over the sampling period 1989-2009 (Sproles et al., 2013). The model was run at 100-m spatial resolution on a daily time step. Because 1 April has historically been the date that water managers have used to represent peak SWE (Stewart et al., 2004; Serreze et al., 1999) we used that SWE data from that date as the predicted variable in the BRT model.  Sproles et al., (2013) showed that 2009 was considered an average snow year so we used 1 April 2009 as our reference year (averaged over 5 days centered

on 1 April). We also used the Sproles et al., (2013) SWE data for the +2°C conditions to represent the spatial distribution of snow for a future average year snowpack.

### 2.3 Analysis

        All spatial data were converted to the same projection and spatial resolution: NAD83, UTM Zone 10, and a 100-m grid cell size. Spatial data were processed using ArcGIS 10.1 (ESRI, Redlands, CA). The "snowpack

bulk" across the MRB was defined as all cells with SWE values within one standard deviation of the basin-wide mean SWE. The area of the snowpack bulk holds the majority of the snow-water volume across the basin. The locations of the SNOTEL sites in the MRB were evaluated relative to the present day and future area of the snowpack bulk for 1 April SWE.





A BRT model was developed to characterize modeled SWE variability across the MRB based on independent physiographic variables using the Classified and Regression Trees (CART) software (Salford Systems, San Diego, CA). The BRT model is a hierarchical non-parametric statistical model that characterizes the mean and variance of a dependent variable using a suite of independent explanatory variables. Modeled SWE and physiographic variable data were used as input data for each cell where snow was present on 01 April 2009. An optimal tree was produced to minimize the standard error of the model, which was then pruned down to the simplest tree possible within one standard error of the optimal tree. The resultant tree identified 20 terminal nodes that characterized the spatial variability in SWE through combinations of independent drivers into 20 BRT-derived snow classes (Table 1). The final BRT model was validated using data for all variables from an independent set of 10,000 randomly selected grid cells from within the MRB.

Using a Geographic Information Systems (GIS) geospatial model and statistically-derived parameters, the 20 BRT-derived snow classes were spatially distributed across the MRB. The geospatial model used physiographic data to distribute the areal extent of each BRT class across the MRB by assigning cells that met the statistically-derived criteria for each BRT class.  Because the BRT-model did not determine a lower elevation limit on snow extent, we excluded areas with an elevation less than 600 m to prevent over-prediction of snow-covered area (SCA). Total volumetric SWE (SWE depth × area) was calculated for each BRT class across the watershed, using the mean and variance of SWE, and the spatial extent of each BRT class. Total volumetric SWE, mean SWE, and the coefficient of variation for each BRT class was used to evaluate the magnitude and spatial variability of BRT-derived SWE estimates relative to modeled estimates for a future average snow year for 1 April 2012 (averaged over five days centered on 1 April).

To create set of feasible locations for the *in situ* snow monitoring network we evaluated the accessibility of locations within the MRB. Using a GIS-based binary selection model, we masked out all private lands and those public lands the presence of endangered Northern Spotted Owl prevented permitted access. We also identified areas within 500-m of a snowmobile-accessible road. From these areas, the final sites were then randomly selected from each of the dominant BRT-derived snow classes within the seasonal snow zone.

## 3 Results

Modeled SWE for 1 April 2009 was normally distributed across the range of elevations throughout the MRB, with the greatest volume of snow located in the mostly forested area between 1300 and 1500 m in elevation (Figure 3).  The four SNOTEL sites in the MRB were located within the area of the snowpack bulk





under current climate conditions (Figure 2a). However, under +2°C conditions, the overall SCA, as well as the area of the snowpack bulk, increased in elevation out of the range of the SNOTEL network (Figure 2b). The area of the current snowpack bulk ranged from 843–1845 m in elevation and contained SWE values from 0.41-0.87 m. The area of the future snowpack bulk ranged from 986 – 1866 m in elevation and contained SWE values from

0.2–0.8 m. Of the 1510 km$^2$ of SCA present on 01 April 2009, 16% was below the lowest SNOTEL site (below 1140 m), 54% of the SCA was within the elevational range sampled by the four SNOTEL sites (between 1140 and 1510 m), and 40% was above the highest SNOTEL site (above 1510 m) in the MRB.

The final BRT model identified elevation, land cover, percent canopy cover, slope, NDVI, and latitude as significant explanatory drivers of the spatial variability of SWE (all selected variables had p-values < 0.05 and

are listed above in order of significance). These explanatory variables characterized SWE across the MRB into 20 distinct snow classes (final BRT model; R$^2$ = 0.95, p-value < 0.01, RMSE = 0.11). Elevation explained the most variance in modeled SWE across the basin, and is the primary driver of all snow classes. In the lower and middle elevations, the BRT model also distinguished snow classes into forested and open/clear-cut vegetation types (Figure 4). Latitude above or below 44.0537° was separated into two classes within the elevation range from

1426–1545 m, however these classes were lumped in the final analysis because we believed the topography of the Three Sisters Mountains in the southeast portion of modeling domain was skewing the statistical distinction of latitude in this analysis. Aspect was not identified as a significant variable driving snow accumulation.

The BRT-derived volumetric SWE estimates had a similar distribution across the elevational gradient as the SnowModel-derived SWE data in the MRB (Figure 5). The BRT-derived estimate of 1.05 km$^3$ total SWE

stored in the snowpack on 01 April 2009 within the MRB was 5% greater than the SnowModel-derived estimate of 0.99 km$^3$. The BRT-derived SCA over-predicted the extent of SWE by 64% compared with SnowModel-derived SCA across the MRB. However 90% of this error was concentrated in the two lowest elevation snow classes. Increasing elevation increased snowpack accumulation, resulting in a greater mean SWE per unit area at the highest elevations. Although these areas only cover a small aerial extent of the MRB, which resulted in

decreasing contribution of total basin-wide SWE above 1791 m. The BRT model performed well across the low, mid, and high elevations. At the lowest elevations, below the snowpack bulk (600–842 m), the BRT-model over-predicted mean SWE by approximately 10%, with a volumetric estimate of 0.054 km$^3$, as compared to 0.049 km$^3$ of SWE as estimated by SnowModel. At the mid-elevations, across the area of the snowpack bulk (843–1845 m), the BRT-model estimated 0.48 km$^3$ of SWE, 11% greater than the SnowModel-derived estimate of 0.43 km$^3$

SWE. At the highest elevations above the snowpack bulk (> 1845 m), the BRT-model estimated 0.516 km$^3$ of SWE, 0.08% less than the SnowModel-derived estimate of 0.52 km$^3$. The low and mid-elevations which consist



of a patchwork of forest harvest and fire disturbance were the areas with the greatest error in the model. Whereas the high elevations above tree line, were the areas with very low error in the model. For a future average year (1 April 2012), the total SnowModel-derived SWE volume across the MRB was 1.64 km$^3$, 50% greater than the total BRT-derived estimate. Due to the high inter-annual variability in SWE, it is not surprising that the basin-

wide BRT-derived SWE estimate showed poor agreement with the SnowModel-derived estimate. However the relative differences in SWE volume and spatial variability between the forested and open land cover types in the 2009 and 2012 SnowModel-derived estimates were captured well by the BRT model at low, mid, and high elevations (Table 2). SWE volume was consistently less in forests compared to open land covers at low, mid, and high elevations in both BRT-derived and SnowModel-derived estimates for 2009 and 2012. The spatial

variability (coefficient of variation) was consistently greater in forests than open areas at the lowest elevations, but relatively similar at the mid and high elevations, in both BRT-derived and SnowModel-derived estimates for 2009 and 2012.

The geospatial selection model identified 16 of the 20 classes as being accessible during winter. The highest elevations in the MRB are far from winter-accessible roads and difficult to monitor due to steep and

avalanche prone slopes. Within the area covered by these 16 classes, random site locations were selected within the six most abundant classes across the MRB to capture low, medium, and high elevations, with forested and open land cover classes. The resultant Forest Elevation Snow Transect (ForEST) monitoring network site locations were thus objectively selected to sample across the range of spatial variability in SWE. The ForEST network, composed of six meteorological stations and snow survey transects, was deployed in November 2011,

and continues to provide high quality snow and climate data to evaluate snow-forest-climate interactions in the MRB (Figure 4).

The ForEST network is unique in that the monitoring site locations were selected based on statistical classification and geospatial analysis, rather than subjective methods that may incorporate bias. The paired forest-open land cover site selection process alone is not unusual, and has already led to important understanding of key

sub-canopy snow processes (Storck et al., 2002; Golding and Swanson, 1986) but here, it has been further validated with the BRT model. The inter-annual consistency in patterns of snow surface energy budget and snow-vegetation interactions across the elevational gradient of the ForEST network suggest that the data are representative of key snow accumulation and ablation processes in the MRB (Figure 6).



# 4 Discussion

As a result of warming winter temperatures, mountain snowpack in the western US will likely continue to decline with potential impacts to forest health (Albright and Peterson, 2013) and streamflow (Jung and Chang, 2011; Cayan et al., 2010), as well as snow-related recreation and tourism (Gilaberte-Búrdalo et al., 2014; Nolin and Daly, 2006). There remains uncertainty around the magnitude of these impacts (Warren et al., 2011; Maurer, 2007; Xu et al., 2005) thus, it is important that monitoring networks not only capture normal snowpack conditions, but capture the range of variability in SWE across the landscape and through time. The SNOTEL sites within the MRB are located within the present day area of snowpack bulk for 1 April SWE, but do not capture the spatial variability of SWE associated with topography and forest cover. The current snow monitoring network was designed based on a historical climate that is not likely to represent future average conditions, it is therefore imperative to evaluate the distribution of SNOTEL sites and consider modification to the network.

Pacific Northwest forests play key role in affecting snow accumulation and ablation across multiple scales however, most research has been conducted at fine scales (Storck et al., 2002) or in areas with cold-dry continental snowpacks (Ellis et al., 2013; Pomeroy et al., 2012). This study emphasizes the watershed-scale control that vegetation and particularly land cover change relative to timber harvest (and potentially fire disturbance) has on snowpack accumulation in the maritime western Oregon Cascades. Understanding the forest structure effects on snow accumulation and ablation across elevation gradients is increasingly important to help guide decision making by local and regional water and forest managers in response to a changing climate.

We developed a snow monitoring network representative of the spatial variability of SWE relative to physiographic landscape characteristics across the MRB; using a coupled BRT statistical classification model, a spatially distributed physically-based SnowModel, and a geospatial selection model. This objective method is a useful tool in determining representative locations for intelligent snowpack monitoring particularly in physiographically complex landscapes. The method of site selection does incorporate uncertainty as a result of compounding statistically-, physically-, and spatially-based models; however, it meets assumptions of non-parametric data analysis, is performed with relative ease, and if data are available for the research basin of interest, it can be well validated. This method could be improved by including more years of input data to fully capture the inter-annual temporal variability in the spatial distribution of SWE.

The ForEST network contributes to the existing SNOTEL network to explicitly investigate snow-vegetation-climate interactions across the range of elevations and forest types in the watershed. After five consecutive years of snow monitoring, we have created a valuable and detailed dataset of snow accumulation,



snow ablation, and snowpack energy balance that spans the spatial variability in forest and open land cover types across an elevational gradient.

## 5 Conclusions

This BRT model characterized peak SWE conditions in an average year, and provided spatially-distributed SWE volume based on physiographic landscape characteristics. This integrated approach informed the distribution of an objective and representative monitoring network that spans the spatial variability in the seasonal snowpack across the MRB (Figure 4).

By quantifying the spatial variability in the key drivers of natural resource distribution, researchers can focus on sensitive areas which may not be identified through traditional site selection means. The use of validated model outputs as a predictor of the spatial variability in snow-vegetation interactions is not new (Randin et al., 2014). The novelty of this research stems from the coupling of a traditional BRT classification process, with a validated physically-based spatially distributed model, to drive a site selection process by its principle parameters across a physiographically complex landscape.

As the scientific community turns to more complex, rescaled, and nested parameterized models to predict ecosystem responses to change, there is still a place for simple approaches to inform scientific research priorities. The uncertainty propagated in nesting multiple models justifies caution in implementing these estimates in management decisions. However in the rugged and densely forested mountain regions of the western Cascade Mountains where there are few alternatives to modeling spatially distributed SWE, this approach provides a validated working hypothesis to guide representative and objective snow monitoring efforts.

## 6 Acknowledgements

This research was made possible through funding from the National Science Foundation (EAR-1039192). Thanks are expressed to Eric Sproles, who provided the modeled SWE data for the MRB and to the Willamette National Forest who provided permits for the ForEST network. Additional thanks are expressed to the many interns who helped install and maintain the ForEST network.



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





**8 Tables**

**Table 1.** The binary regression tree (BRT) model characterized SWE within the McKenzie River Basin into 20
snow classes defined by the following physiographic parameters: elevation (m), land cover (forested vs.
open/clearcut), percent forest canopy cover (CC), slope, NDVI, and latitude (Y). The bold lines represent the
BRT snow classes used for the ForEST network of snow monitoring stations which have been continuously
monitoring snow processes in paired forest and open sites at low, medium and high elevations since 2012. The
monitoring site in snow class 15 also collected continuous data for 2012 and 2013 but then was removed due to
logistical and financial restraints.

| Snow Class | Elevation | Veg Class | Other | Snow Class | Elevation | Veg Class | Other |
|---|---|---|---|---|---|---|---|
| 1 | <1121 | Forest | | **11** | **1333-1426** | **Open** | |
| 2 | 1122-1199 | Forest | CC<20% | **12** | **1426-1545** | **Forest** | |
| **3** | **1122-1199** | **Forest** | **CC>20%** | **13** | **1426-1545** | **Open** | |
| 4 | <977 | Open | | 14 | 1546-1791 | | Y<44.1° |
| **5** | **977-1199** | **Open** | **Slope<27°** | **15** | **1546-1791** | | **Y>44.1°** |
| 6 | 977-199 | Open | Slope>27° | 16 | 1792-1919 | | |
| 7 | 1200-1255 | | NDVI<0.2 | 17 | 1920-2039 | | |
| 8 | 1200-1255 | | NDVI>0.2 | 18 | 2040-2371 | | |
| 9 | 1255-1332 | | | 19 | 2372-2788 | | |
| **10** | **1332-1426** | **Forest** | | 20 | >2788 | | |





**Table 2.** Differences in volume (mean), standard deviation (SD), and coefficient of variation (CV) of SWE between forested and open sites (forest minus open) for low, medium, and high elevations (BRT classes where we have installed micrometeorological towers at the same elevation in forest and open sites). Values were derived from the BRT-derived estimates and the 2009 and 2012 SnowModel-derived estimates.

| Forest - Open | Low | | | Medium | | | High | | |
|---|---|---|---|---|---|---|---|---|---|
| | Mean | SD | CV | Mean | SD | CV | Mean | SD | CV |
| BRT | -0.23 | -0.06 | 0.63 | -0.2 | -0.002 | 0.02 | -0.16 | -0.01 | 0.004 |
| 2009 | -0.06 | -0.01 | 0.13 | -0.08 | -0.03 | -0.02 | -0.09 | -0.03 | -0.03 |
| 2012 | 0.01 | -0.01 | 0.05 | -0.006 | -0.01 | 0.03 | -0.11 | 0.02 | 0.02 |





**9 Figures**

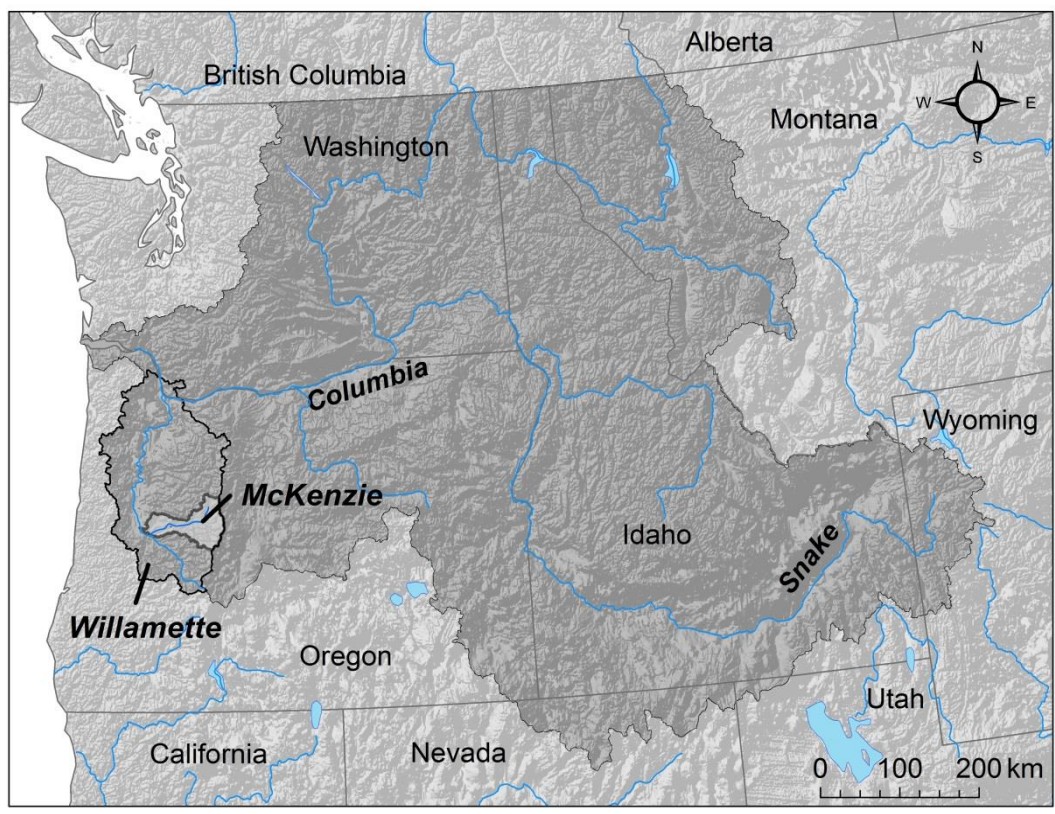

**Figure 1.** McKenzie River Basin is nested in the Willamette River Basin within the greater Columbia River Basin.





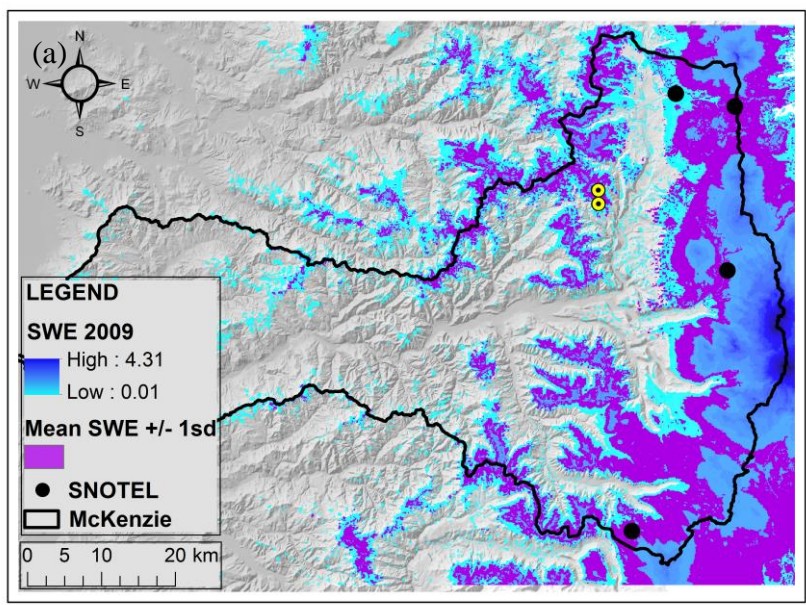

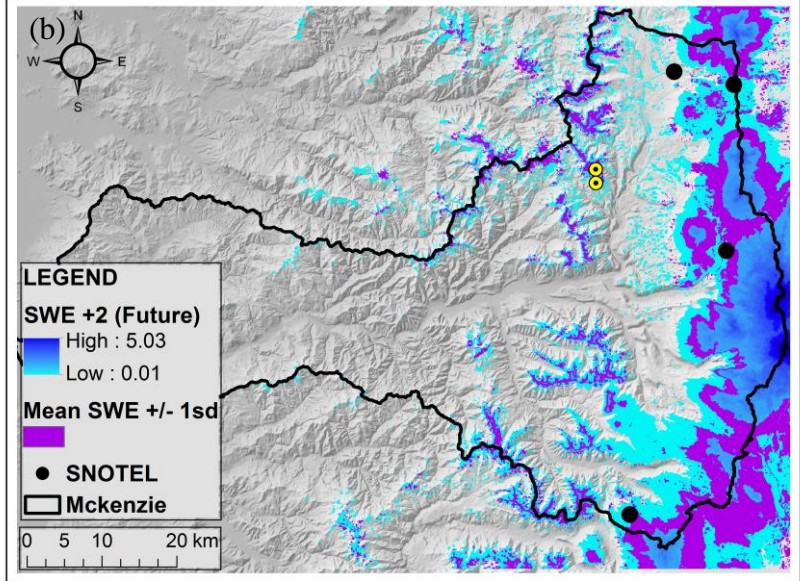

**Figure 2.** SnowModel-derived SWE in shown in blue and the area of the snowpack bulk shown in purple (+/- 1 standard deviation around mean basin-wide SWE) for (a) 1 April 2009 (average snow year) and (b) 1 April for +2° C conditions (future average snow year). Four SNOTEL sites present in the MRB in 2009 (shown in black) are located in the area of the snowpack bulk for 2009, but not under +2° C conditions. Two new SNOTEL sites (shown in yellow) were installed in 2012.





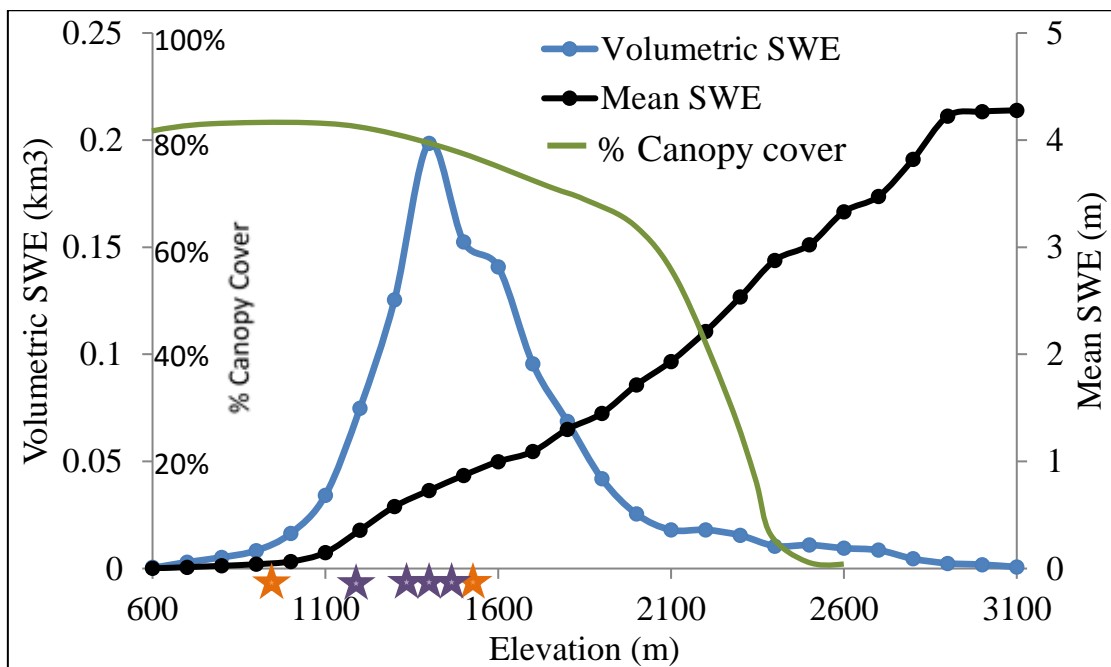

**Figure 3.** Elevation distribution of SnowModel-derived SWE data in the McKenzie River Basin for 01 April 2009. The area of greatest SWE volume persists in a narrow elevation range which is monitored by four historical and two newly installed (as of 2012) SNOTEL stations (elevations of historical stations shown as as purple stars and new stations as orange stars).





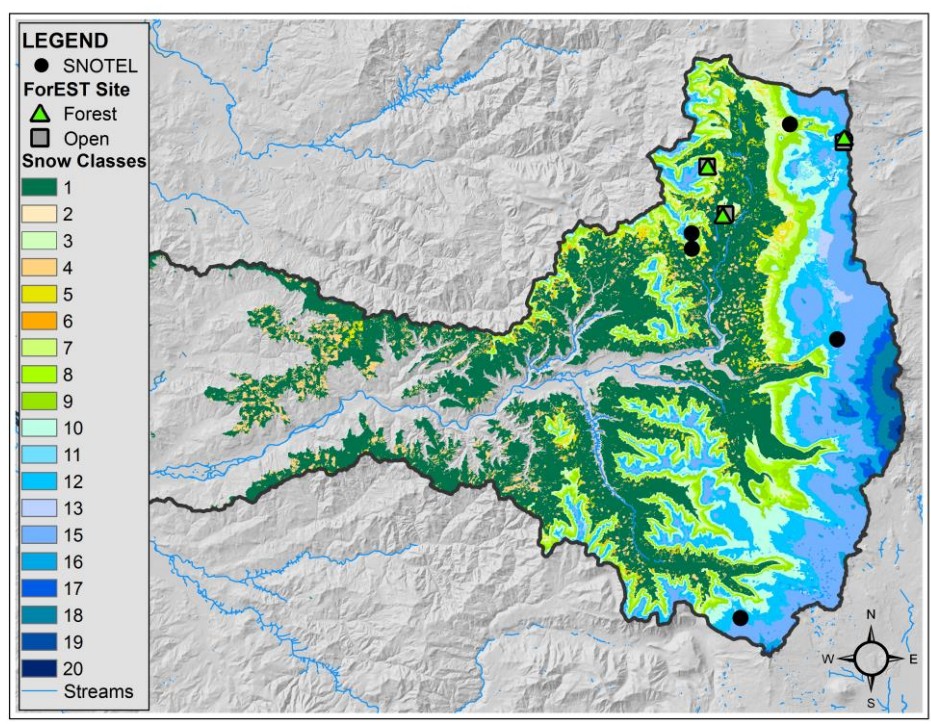

5    **Figure 4.** BRT-derived snow classes distributed across the McKenzie River Basin. The selected locations for the snow monitoring sites were not evenly distributed in space, but were selected to span the range of spatial variability in snow-vegetation-climate interactions.





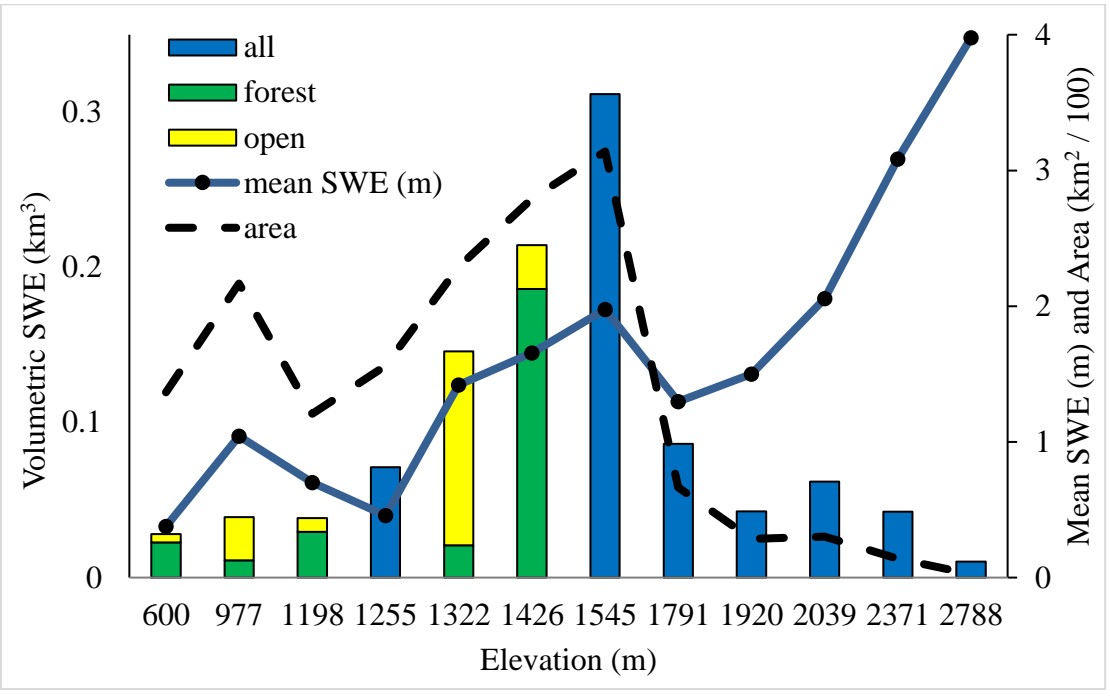

**Figure 5.** BRT-derived volumetric SWE (km³) across the McKenzie River Basin for each BRT class stacked along the elevation range where the classes are located (minimum elevation for each class labeled on x-axis). In the low-, and mid-elevations, the forest vs. open distinction is statistically important in distinguishing snow classes. In the high-elevations, above treeline, only elevation significantly drives variability in snow accumulation. Mean SWE increases but volumetric SWE decreases as the land area decreases at the highest elevations.



**Figure 6.** Mean SWE (cm) from snow course measurements collected at the paired open and forested snow monitoring sites in the ForEST network at (a) high-, (b) mid-, and (c) low-elevations during the winters of 2012, 2013, and 2014. Light blue bars represent mean SWE (cm) in open sites. Green bars represent mean SWE (cm) in forested sites. Error bars indicate the maximum and minimum measured SWE (cm) from 2012, 2013, and 2014.