# Peer review of "Developing a representative snow monitoring network in a forested mountain watershed"

_Hydrology and Earth System Sciences, 2016_

## Referee Comment (RC1) · Anonymous Referee #1 · 7 Aug 2016

Developing a representative snow monitoring network in a forested mountain watershed Kelly E. Gleason, Anne W. Nolin, and Travis R. Roth

The authors present a comparison of a binary regression tree (BRT) statistical model, trained using a distributed snow model (SnowModel), to spatially locate similar snow classes around a watershed which guides the siting of meteorological stations (6 stations at three sites). Two snapshots of spatial snow distribution are used: 2009 (training data) and 2012 (evaluation data) in order to evaluate the BRT and demonstrate its utility for met station siting. This concludes with the claims that it improves the basis for site selection over a physically based model due to the uncertainty propagated by

parameter selection (i.e. nested sub-models) in physically-based models.

As the manuscript is currently written, there are some substantial issues to respond to as well as a few minor suggestions:

1. I don't see how this is novel science from the perspective of BRT applications. The authors provide six citations in the introduction to similar BRT work and explicitly mention in their conclusions that it is not an advance over Randin et al. (2014).

2. This work demonstrates that a statistical BRT model that is not temporally responsive to a warming climate (i.e. in the same way that SNOTEL data provide temporally static statistical relationships to discharge), performs worse than the distributed physically-based model (SnowModel). Table 2 shows this performance difference is by an order of magnitude in the mean values for medium and low elevations. Hence the assertion in the conclusions that there is still a place for simple approaches is undermined. From the presented methodology of the BRT model it seems this is not a simple approach, and in a watershed where a physically-based model can (and has) been deployed, it offers no improvement. While there may be uncertainty in many parameterizations and process representations of physically based models, at least they will be responsive in outputs to changing input in a warming climate (especially relevant to the pacific north-west region).

3. The claim of a predictive system (whether BRT or a physically based model) as a tool for advancing the siting of met stations is very site specific and doesn't provide wider scientific advancement. Local watershed knowledge of potential site access, elevation and forest/open areas would likely provide just as much information required as a complex statistical BRT style analysis. While this style of statistical analysis may have been useful to justify the location of met sites in the MRB watershed, in itself, it doesn't justify either a methodological or scientific advance in HESS.

4. The benefits of a BRT approach remain poorly quantified. In the abstract, elevation, vegetation type and vegetation density are defined as the significant drivers of

SWE distribution. As we already know this is important in montane environments this does not come as a surprise, however, not providing any statistical quantification of the relative significance (nor on the main body of text) means such a major concluding statement adds little to the current body of work in the literature.

Minor comments:

Abstract: this could be condensed substantially. Ln 9-14 and 24-27 could be shortened/removed. No quantified results are presented. The reader is left unaware how representative (i.e. quantified) this BRT model actually is.

Pg 1, Ln27: The idea this paper tests the MCB snow network within a projected warming climate (from 2009 to 2012) suggests something that is not adequately delivered by this paper.

Pg 4, Ln 7 & 19 – don't need 'Description of the' in either sub heading.

Pg4, ln 12 – 'which' is grammatically correct after a comma rather than 'that'. Pg 4, Ln 25-27 – following Winstral et al., (2002) and subsequent papers by Winstral et al., was this used to calculate redistribution of snow (especially above tree line) in drifts which are very important hydrological areas to get SWE correct in a watershed?

Pg 5, Ln 18-21 – while Sproles et al. (2013) is often cited, as this is such a key foundation to this work it needs greater explanation in this paper – in particular how the future SWE conditions are calculated, and especially the change to precipitation rates and phase (rain/snow) as well as temperature.

Pg 5, ln 24 – can more be said about issues of up-scaling (aggregation) and downscaling (disaggregation) of different data sets?

Pg5, Ln 25 – why concentrate on areas defined as 'bulk' rather than fully spatially distributed models? Locating big drifts, often above tree line, are key to understanding the timing and magnitude of discharge. This seems to have been neglected under this BRT model.

Pg 6, Ln 5-10 – The way that SnowModel is combined or used to evaluate BRT is presented in a very confusing fashion. Where is the independent data to evaluate BRT?

Pg 6, ln 12 – 20 BRT snow classes? Wasn't one removed due to logistics and finance? This adds confusion to the methods.

Pg 6, ln 14-16 – Why were lower elevation extents removed? This is done without any quantification nor real justification.

Pg 6, ln 16-17 – what proportion of the basin was removed? Why do this if it is a SWE contributing area to discharge, why would this cause over prediction?

Pg 6, ln 21 – add 'a' between 'create' and 'set'.

Pg 6, ln 24 – why is 500m threshold applied? In practice one would expect field locations for met sites to be closer or further away from transport links depending on local conditions (i.e. how potential met site locations have always previously been evaluated).

Pg 7, Ln 8 – the 'final' BRT model. How many BRT models were evaluated? The rest of this paragraph has already been discussed and is providing repetition.

Pg 7, Ln 14 – why does latitude matter?

Pg7, Ln 15 – why does aspect not matter? Especially for snowmelt rates, this goes against conventional wisdom.

Pg 7, Ln 18 – why were BRT and SnowModel not used in conjunction with each other. When both are available it is confusing that they are not used together to optimize estimation of SWE distribution.

Pg 7, Ln 20 – BRT estimation of mass should be good in 2009 as it is tuned with SnowModel, but poor prediction of SCA (64% SCA over prediction) suggests it's not getting SWE right for the right spatial reasons (i.e. at low elevation).

[Figure]

Pg 7, Ln 23 – Increasing elevation does not increase accumulation, it is increases with elevation (i.e. not a cause in itself).

Pg7, Ln 26-31 – could this information be put into a table?

Pg 8, Ln 1 – comma needed after 'Whereas'.

Pg 8, ln 4-5 – How does BRT adapt to changes in winter precipitation inter-annually? If it can't, what advantages does it have over running SnowModel?

Pg 8, Ln 7 – SnowModel derived estimates were NOT captured well by BRT. They were an order of magnitude different at low and medium elevations. Need a much better quantified argument to justify this.

Pg 8, Ln 13 – Need to provide more about how accessibility is determined as a criteria.

Pg 8, ln 19 – six met stations is a bit misleading, rather there are three sites, each with adjacent open/forest met stations.

Pg 8, Ln 17-26 – this isn't a scientific result unless you then go on to do something with these met data.

Pg 8, Ln 26 – how has this been stringently validated with the BRT model?

Pg 8, Ln 26-28 – Consistency in the pattern of measured snow course SWE doesn't corroborate energy balance and snow-veg interactions.

Pg 9, Ln 15-16 – This study doesn't explicitly demonstrate the impact of timber harvest / fire disturbance impact on SWE distribution.

Pg 9, Ln 20-21 – If BRT and SnowModel are coupled (as stated) then what does this combination give us that SnowModel doesn't give us as a stand-alone product? This is not providing added information on hydrological response units (HRU), it is not a new idea in snow hydrology (e.g. CRHM), and doesn't provide an obvious robust advancement in inter-annual transferability.

Pg 9, Ln 26 – Yes, inter-annually transferability really needs to be more robustly tested by this methodology, rather than one 1 April snapshot in 2012. Currently this evaluation/validation has not been sufficiently done with independent data.

Table 1 – What percentage of SCA was above 1546m (was it ∼40%)? If these data were rejected can this be demonstrated that this is not a problem? While thin SWE and scour is likely in Alpine areas above tree line drifts in these areas can contribute substantially to the timing of increased discharge through melt-out.

Table 2 – no units. Can low, medium and high be classified? Which sites were the forest and open sites – can these be related to a map or specifically described?

Fig 2 – put yellow circles in legend. Cite Sproles in caption (see previous comment about more explicit explanation of future precipitation scenario in Sproles data).

Fig 3 – I am surprised that mean SWE by elevation increased above tree-line, would have expected some thinning of SWE due to scour, can this be explained? The hypsometry of the basin would be a very useful (essential?) addition to this figure.

Fig 4 – relate snow classes to the Table otherwise they make no sense.

Fig 5 – How does forest and open relate to the 'all' classification? What is additional to 'all' other than forest and open? Why is mean SWE so different to SnowModel? Which year is this for? Don't put descriptive results in caption, put them in the main body of the text. Caption says it's statistically important, where is this statistical analysis?

Fig 6 – This is just measured SWE, how is it use to quantitatively evaluate the new modelling framework? Need to define the high, mid and low elevations in the caption. Error bars seem to be the range rather than any calculation of error.

---

## Referee Comment (RC2) · Anonymous Referee #2 · 15 Sep 2016

General Comments

Gleason et al. present a case study detailing how a binary regression tree (BRT) model can be used to identify major statistical classes of snow accumulation based on readily observable physiography. This regression tree classification/ model subsequently is used to inform a detailed snowpack monitoring array/ network. The specific application is in the MacKenzie River basin in the Cascade Mountains of Oregon, an area subject to strong elevation gradients in snowpack accumulation and potential quite susceptible to climate change. The result of this effort was the establishment of the Forest Elevational Snow Transect (ForEST) which has been operational for five years.

The case is made that an objective approach like the BRT is preferable to errors as-

sociated with a subjective siting based on heuristics or experience. Specifically, the drivers are: elevation, land cover, percent canopy cover, slope, NDVI, and latitude. Not terribly surprising, but nice to see objectively defined. Although technically sound, the paper could do a better job of closing the circle on a compelling story or novel contribution. Discussion of climate change, monitoring, and site selection are mixed making it difficult to determine exactly what is the contribution of this effort beyond adding another site at slightly higher elevation. A clearer analysis of how the BRT guided siting resulting in a more representative/ predictive/ useful network in a climate change scenario is needed. For example, how can others use this approach without physically validated SWE distributions? Should we expect this classification to hold elsewhere or even here under climate change? Indeed, should we move the discussion beyond April 1st to better inform water resource management? That date is a compromise after all and it very well may be that a more representative/ useful for water resources monitoring network is not wedded to that date (heresy I know). I can't say which of the above (or others) are most worthwhile avenues to pursue, but after reading the manuscript several times, I am left with the sense that something is missing.

Specific Comments

Although there is a certain level of objectivity in the BRT, I suggest that the modeling approach carries its own set of biases first in the physical model which must scale point observations to a 3-D snow cover while only the wind redistribution part of the model operates in 3-D including non-local effects. A number of papers suggest that edge effects of vegetation on energy balance as well as remote topographical vs local vegetation shading influence snowpack mass and energy balance. These are difficult to include, especially at the scale of this exercise, but they clearly may be relevant for future scenarios. Instead, the strength (of using the BRT and physical model) is being able to accurately evaluate these assumptions by making them explicit, rather than suggesting they don't exist. No model is perfect.

Second, the statistical approach used in the BRT assumes stationairity in the processes from year to year for your comparisons as well as for future predictions under climate change. This leaves me confused by the 50% difference in BRT vs physical model SWE for 2012. How was the BRT initialized for 2012? Did you force the tree structure to replicate that from 2009 or did you let the model form its own structure? The observation that patterns persist is nice, but it seems that there is information in the differences between the two estimates of SWE volume.

I suggest showing the tree and order/ strength of nodes. This would strengthen the presentation of the BRT relative to the table.

Figure 3 – At first glance (and second) it seems that only one of the new locations is in the area of volumetric SWE accumulation. In other words, one location is too low in elevation and there is a large volume higher in elevation that your highest site. Just eyeballing it perhaps 30 to 40% of SWE is above that location.

Figure 5 - I find this confusing - the classes open, forest, and all don't seem to make sense. perhaps you are trying to communicate too many messages? area, volume, and controls all in one figure with no indication that the volume differences between forest and open are due to area or depth

Figure 6 why show max and minimum as it muddies the difference between location and year.

Technical comments

Page 1, Lines 22-26 – It seems that the current ForEST network of sites is the result of the BRT modeling exercise, if so, say that directly rather than back in to it as currently written

Page 4, Line 25 and on to next page - Unclear how NLCD, 30m LANDFIRE, and 250m NDVI were used (what data from each source) and/ or aggregated to reach 100m spatial resolution. Specify this up front so the reader doesn't need to go back and forth between results and methods

Land cover is variously referred to as forest or open, and is synonymous with veg class, correct? This should be clarified.

Table 2 – what are units?

[Figure]

---

## Referee Comment (RC3) · Anonymous Referee #3 · 27 Sep 2016

SUMMARY OF THE PAPER

This paper (1) investigates which physiographic factors influence modeled spatial SWE distributions on 1 April in the McKenzie River Basin of the western Oregon Cascades and (2) demonstrates how this knowledge can be used to locate new snow studies sites in an objective way for resolving physiographic influences on SWE. The work is motivated to inform observational network design in snow-dominated watersheds where forest change and climate change present challenges. They use binary regression trees (BRT) to predict 1 April SWE in an average year based on predictors such as elevation, forest cover, NDVI, and latitude. This is a unique application of BRT because they use 1 April SWE output from a spatially distributed physically-based model (Snow-

Model) at the watershed scale, whereas most previous BRT snow studies have been at smaller scales and with observational data. The analysis examines 20 snow classes from BRT in average snow years in current and future (+2 C degree) conditions. The study compares differences in SWE in forest and clearings at different elevations, as sampled in the ForREST network.

I think the most major contribution of the paper is that it demonstrates a method for utilizing physically-based model output to improve observational network design. The method is novel and the results should garner decent interest from the community. I think the writing/figures are of especially high quality. This paper should be published in HESS after addressing a variety of major and minor comments (below).

MAJOR SPECIFIC COMMENTS

1. The most glaring weakness of the analysis is that it does not address collinearity of the predictors anywhere. Certainly some of the predictors in the BRT co-evolve in space. For example, forest cover decreases with elevation (seen in Figure 3). How can one disentangle the unique influence of covarying predictors within the adopted regression framework, let alone assert which predictors dominate SWE (Page 7, Line 12)?

2. The analysis implies that current observational sites may not be representative of snow conditions in a future climate and that physically-based model outputs are valid irrespective of climate conditions. However, SnowModel (and other models with the physically-based label) do have embedded routines/parameterizations that are empirical in nature and tuned to historical conditions (e.g., atmospheric longwave radiation). There is a general lack of discussion about the reliability of models in projecting changes outside of historical conditions. These approximations of the real-world are further muddied here because the study is advancing a model (BRT) of a model (Liston/Elder SnowModel).

3. Comparing basin SWE on 1 April for the current climate and a warmer +2 degree
C climate may be misleading/inappropriate, as the basin may be well into the melt season by 1 April in the warmer climate. 1 April is historically significant only because it has been (on a mean basis) near peak SWE timing. Arguably, the date of peak SWE will advance earlier in the year with climate warming. So analyzing 1 April in a future warmer climate is like analyzing a date in mid- or late- April in the current climate, and we might say that SNOTEL sites are unrepresentative of basin conditions once melt conditions have advanced to that date in late April. However, that is not a fair comparison, as the SNOTEL sites may have been more representative of mean conditions earlier in the season (i.e., near peak conditions). To address this potential issue, the authors should consider not only the spatial distribution of SWE but also the temporal evolution. Are the SNOTEL sites more representative of basin SWE at an earlier date (e.g., March 15) in the warmer climate?

MINOR SPECIFIC COMMENTS

1. The "Future year (1 April 2012)" terminology versus +2 degree C year terminology is inconsistent and confusing at times. How can April 2012 be "a future year" when it is now (in 2016) well in the past (e.g., Page 6, Lines 19-20)? This needs better explanation. Also, please consider revising the language throughout the manuscript.

2. The "high inter-annual variability in SWE" is offered as a reason for differences in SWE volume from BRT vs. SnowModel in the future scenario (page 8, line 4). However, this does not make sense, given that only average years are considered in the analysis, effectively precluding any influences of inter-annual variability. The authors go on to contradict the above assertion about inter-annual variability in the discussion: "This method could be improved by including more years of input data to fully capture the inter-annual temporal variability in the spatial distribution of SWE." Please revise.

3. Was this analysis actually conducted prior to the installation of the ForEST network in November 2011? Or is this a retrospective analysis to test the representativeness of the established network? The connection between the presented work and the design

of the ForEST network is never really made clear. This distinction has implications for the title and tone of the manuscript. Currently, the manuscript implies that the analysis was used to inform the design of the ForEST network (page 10, lines 5-7). The current title is appropriate if the analysis with April 2009 was conducted first. However, if this is a retrospective analysis of the adequacy of the network, then the title may be better stated as "Testing the representativeness of a snow monitoring network in a forested mountain watershed".

TECHNICAL CORRECTIONS

- Page 2, Line 28: Add "currently" before manages (the number of SNOTEL stations changes in time).

- Page 4, Line 9: "In the heart of" is somewhat colloquial; consider rephrasing this sentence.

- Page 4, Lines 21-22: There is some overlap between these variables and at this point it is unclear how they are uniquely distinguished. For example, incoming solar radiation will vary with slope, aspect, and vegetation, all of which are variables listed here. Is there something unique about "solar radiation" that you should list it here? Does it vary with atmospheric conditions? Please clarify.

- Page 5, Line 16: Presumably the model was run at a sub-daily time step (necessary for physical models), but the model provided outputs on a daily basis. Please rephrase.

- Page 5, Line 24: Please provide more information about how finer resolution spatial data (e.g., 10-m elevation, 30-m land cover data, etc.) were aggregated to 100-m, and how coarser resolution spatial data (e.g., the 250-m NDVI data) were resampled/downscaled to 100-m.

- Page 5, Line 24: You already cited the maker/city of ArcGIS, so I am unsure if you need to do it again.

- Page 5, Line 27: Did you use the publically available locations of the SNOTEL sites?

The publically available coordinates are imprecise.

- Page 6, Lines 2-4: Again, I question the independence of the physiographic predictor variables.

- Page 6, Line 21: Add "a" before "set".

- Page 6, Line 23: Revise to say "and public lands where the presence. . .".

- Page 6, Line 27: Did you test for normality? Perhaps include the skew and kurtosis. There is a bit of a skew toward higher SWE volume at the higher elevations, which is why I ask.

- Page 7, Lines 1-2: Consider including a separate SWE volume line in Figure 3 for the climate change scenario. This will provide another way of showing the shift toward higher elevations above the SNOTEL sites (in addition to the spatial plots in Figure 2).

- Page 7, Line 3: Is this SWE range measured or modeled at the SNOTEL sites? Please state.

- Page 7, Line 11: Please include units on the RMSE.

- Page 7, Line 12: How much variance did elevation explain? Please quantify.

- Page 7, Line 15: Recommend using a different word than "believed". Also, it is possible to test the influence of the Three Sisters – just exclude those points in the BRT anaylsis and compare the resulting regression trees.

- Page 7, Line 20: Should this be ~6%? 1.05/0.99 = 1.061 or 6.1%.

- Page 7, Lines 24-25: Check the sentence: "Although these areas. . .. Above 1791 m." This does not appear to be a complete sentence.

- Page 8, Line 1: Please clarify which model when you state "greatest error in the model". I think it is the BRT model. Also, the use of the term "error" implies that the SnowModel output is "truth" in the comparison, which may be tenuous. Consider using
some language like "difference between models" in this context.

- Page 8, Lines 22-28: This is more appropriate for the discussion section, not the results section.

- Page 9, Line 12: Add "a" before "key role".

- Page 9, Line 20: Improper semi-colon usage. You can safely remove it, or break the sentence into two here.

- Page 9, Line 23: Replace "does incorporate" with "incorporates".

- Page 9, Lines 23-26: This is a long and overly complicated sentence. Please rephrase and/or revise into shorter sentences.

- Page 10, Line 19: If a hypothesis is validated, is it still a "working hypothesis"? The word choice is puzzling here.

TABLE AND FIGURE COMMENTS

- Figure 2 caption: Replace "in shown" with "is shown".

- Figure 2 caption: Please define the units of SWE.

- Figure 2: If April 2009 is an average year (page 5, line 19) and the climate change scenario is a 2 degree C perturbation to an average year, why is the maximum SWE lower in April 2009 (4.31) than in the climate change scenario (5.03)?

- Table 1: What is the logic of the organization of snow classes in Table 1? It generally goes from low to high elevation, except the 977 to 1199 elevations are not in order. Please rectify.

- Table 1: Should snow class 1 read "977-1199" instead of "977-199"?

- Table 1: Consider showing statistics with each snow class to record how well the regression works in that group.
- Table 1: What is the purpose of having a binary vegetation class (forest vs. open) and forest canopy cover (CC) predictor variables? Would it not be more straightforward to just include CC and let the BRT tell us when/where the binary distinction dominates the SWE response?

- Table 1: In some (but not all) cases, there is an overlap in the elevation. Is a location at 1426 m elevation in the open in snow class 11 or snow class 13?

- Figure 3: Please use a superscript for cubic km on the left y-axis.

- Table 2: It is unconventional to have negative standard deviation or coefficient of variation. Please make these positive. Also, are the CV numbers correct? They should be the SD/Mean, but that does not appear to be the case here.

- Table 2 caption: Please include the units of SWE differences here.

---

## Author Comment (AC1) · 5 Nov 2016

**Dear Reviewer,**

Thank you for your comments and recommendations for the revised manuscript, they were very helpful in presenting this research in a more robust and defensible way. In order to tell a more compelling story, we have made multiple changes to the revised manuscript. We focused the paper solely on the objective approach to improve snow observational network design, and therefore omitted the evaluation of the SNOTEL network under climate change. We acknowledge the limitation in the initial analysis conducted in 2010 which was based on data from 01 April 2009, with the assumption it represented maximum snow accumulation across the basin during an average snow

year. To improve upon this in the revised manuscript we used data from the five days centered on the date of actual peak SWE in the McKenzie River Basin for an average year 2009, an above average year 2008, and a below average year 2005. Evaluating the BRT-derived snow classes from three years of SWE data enabled us to use a more robust analytical approach including omission and commission statistics of overall classification accuracy.
The authors present a comparison of a binary regression tree (BRT) statistical model, trained using a distributed snow model (SnowModel), to spatially locate similar snow classes around a watershed which guides the siting of meteorological stations (6 stations at three sites). Two snapshots of spatial snow distribution are used: 2009 (training data) and 2012 (evaluation data) in order to evaluate the BRT and demonstrate its utility for met station siting. This concludes with the claims that it improves the basis for site selection over a physically based model due to the uncertainty propagated by parameter selection (i.e. nested sub-models) in physically-based models. As the manuscript is currently written, there are some substantial issues to respond to as well as a few minor suggestions:

Your comments highlight the need to clarify a few key points in the revised manuscript that may have been initially misinterpreted. For example we conclude that the presented method of site selection is an improvement over more commonly used heuristic approaches, but because the method couples physically-based, statistically-based, and geospatial models there is uncertainty particularly in predicting future conditions. Here is the revised paragraph in the discussion section which addresses your comment above, "We developed a snow monitoring network representative of the spatial variability of SWE relative to physiographic landscape characteristics across the MRB for an average, above average, and below average snow year; using a coupled BRT
statistical classification model, a spatially distributed physically-based SnowModel, and a geospatial selection model. This objective method is a useful tool in classifying snow characteristics across the landscape to determine representative locations for intelligent snowpack monitoring particularly in physiographically complex landscapes. Although it is an improvement over more commonly used heuristic approaches to site selection, the method incorporates uncertainty as a result of compounding statistically-, physically-, and spatially-based models which justifies caution in implementing these estimates in management decisions. However, the method meets assumptions of nonparametric data analysis, is performed with relative ease, and if data are available for the research basin of interest, it can be well validated. As even physically-based models incorporate inherent empirically-based historically-derived assumptions, there is also uncertainty in using this approach to represent future spatial variability in snow accumulation."

1. I don't see how this is novel science from the perspective of BRT applications. The authors provide six citations in the introduction to similar BRT work and explicitly mention in their conclusions that it is not an advance over Randin et al. (2014).

We present a novel method in designing an objective and representative snow monitoring network, which promotes the opportunity for novel science. In the introduction we acknowledge research which has used the BRT to evaluate snow accumulation at small scales, or snow covered area at broad scales, however no previous work has coupled physically-based model output with non-parametric statistical models to improve snow monitoring network design. In the introduction we include the following paragraph explaining how this method goes beyond any previous work using BRT modeling, "Landscape characteristics have been used to predict snowpack conditions at hillslope scales using non-parametric binary regression tree (BRT) statistical classification models (Molotch et al., 2005; Anderton et al., 2004; Erxleben et al., 2002; Winstral et al., 2002; Balk and Elder, 2000; Elder et al., 1998). Larger scale BRT approaches have also been conducted using remotely sensed snow-covered area and
interpolation methods (Molotch and Meromy, 2014; Molotch and Bales, 2006b). However, no study to date has used landscape characteristics in conjunction with modelled and validated physically-based and spatially distributed SWE data to understand physiographic drivers of snow accumulation at broad scales (watersheds > 1000 km2) or to identify optimal locations for snowpack monitoring. Additionally, most of the research on the physiographic relationships to snow processes has been done in cold-dry continental snowpacks where mid-winter melt events are infrequent and wind redistribution is substantial (Molotch et al., 2005; Erxleben et al., 2002; Winstral et al., 2002; Balk and Elder, 2000). Much less is known about how physiographic conditions influence the temperature sensitive snowpacks in the forested maritime basins of the Pacific Northwest."

Also in the conclusions we include the following paragraph, which specifically describes the novelty of this research," By quantifying the spatial variability in the key drivers of natural resource distribution, researchers can focus on sensitive areas which may not be identified through traditional site selection means. The use of validated model outputs as a predictor of the spatial variability in snow-vegetation interactions is not new (Randin et al., 2014). The novelty of this research stems from the application of the method, where by the coupling of a traditional BRT classification process with a validated physically-based spatially distributed model, we improved observational network design in a forested montane watershed."

2. This work demonstrates that a statistical BRT model that is not temporally responsive to a warming climate (i.e. in the same way that SNOTEL data provide temporally static statistical relationships to discharge), performs worse than the distributed physically-based model (SnowModel). Table 2 shows this performance difference is by an order of magnitude in the mean values for medium and low elevations. Hence the assertion in the conclusions that there is still a place for simple approaches is undermined. From the presented methodology of the BRT model it seems this is not a simple approach, and in a watershed where a physically-based model can (and has)
been deployed, it offers no improvement. While there may be uncertainty in many parameterizations and process representations of physically based models, at least they will be responsive in outputs to changing input in a warming climate (especially relevant to the pacific north-west region).

To simplify the manuscript and focus on the novelty of our method, we have removed the evaluation of the SNOTEL network under a warming climate. Also, in order to make the validation of this approach more robust in the revised manuscript, we have included three years of input data and allowed the BRT model to build its own structure from an average snow year, an above average snow year, and a below average snow year. By including these additional years, we were able to use omission vs commission statistics to determine the overall accuracy of the models between years of current snow conditions. We hope this clarifies some of the confusion mentioned in the above comment. We do not suggest that a BRT modeling approach is more robust than a physically-based model in predicting snow volume across a watershed. We present a relatively simple method using coupled models to classify the snowpack (a normally continuous variable) across a complex watershed to guide objective snow monitoring network design. We have also included the following statement in the discussion, "As even physically-based models incorporate inherent empirically-based historically-derived assumptions, there is also uncertainty in using this approach to represent future spatial variability in snow accumulation."

3. The claim of a predictive system (whether BRT or a physically based model) as a tool for advancing the siting of met stations is very site specific and doesn't provide wider scientific advancement. Local watershed knowledge of potential site access, elevation and forest/open areas would likely provide just as much information required as a complex statistical BRT style analysis. While this style of statistical analysis may have been useful to justify the location of met sites in the MRB watershed, in itself, it doesn't justify either a methodological or scientific advance in HESS.

Using an objective method of site selection is rarely used, and we suggest is an ad-

**HESSD**
vancement over more heuristic approaches which require institutional knowledge that may not exist in remote rugged watersheds. Although the result of this analysis may not be particularly surprising, it is a useful method for objectively validating our assumptions about "representativeness" of any particular monitoring site location. For example, our final station locations may appear clustered in physical space, but from using our method we are confident the locations span the parameter space of the key drivers influencing the spatial variability of snow accumulation across the watershed. We present this method with the hope that more scientists will objectively distribute future monitoring locations based on actual data instead of going on "gut feeling".

4. The benefits of a BRT approach remain poorly quantified. In the abstract, elevation, vegetation type and vegetation density are defined as the significant drivers of SWE distribution. As we already know this is important in montane environments this does not come as a surprise, however, not providing any statistical quantification of the relative significance (nor on the main body of text) means such a major concluding statement adds little to the current body of work in the literature. We hope we have clarified in the final manuscript the benefits of this coupled approach to objective site selection. As stated above, the results of this analysis are not surprising and validate already known assumptions about snow-vegetation interactions in montane watersheds. However what is novel in this method is that it uses statistically derived relationships to classify the spatial distribution of snow by its primary drivers to improve observational network design. Also, because we included three years of input data in the final manuscript, we include a more robust statistical validation of the BRT model between years.

Minor comments:

Abstract: this could be condensed substantially. Ln 9-14 and 24-27 could be shortened/removed. No quantified results are presented. The reader is left unaware how representative (i.e. quantified) this BRT model actually is.
As suggested, we have condensed the abstract to focus more simply on the method of site selection, and included a basic statistic of the final BRT model in the revised manuscript. The revised abstract now reads, "A challenge in establishing new groundbased stations for monitoring snowpack accumulation is to locate the sites in areas that represent the key processes affecting snow accumulation and ablation. This is especially challenging in forested montane watersheds where the combined effects of terrain, climate, and land cover affect seasonal snowpack. We present a coupled modelling approach used to identify a parsimonious set of monitoring sites in a forested watershed in the western Oregon Cascades mountain range. We used a binary regression tree (BRT) non-parametric statistical model to classify peak SWE based on physiographic landscape characteristics in a normal year, an above average year, and a below average year. Training data for the BRT classification were derived using spatially distributed estimates of SWE from a validated physically-based model of snow evolution. The optimal BRT model showed that elevation and vegetation type were the most significant drivers of SWE in the watershed (R2 = 0.93, p-value

manuscript to focus on the objective method of site selection to capture the spatial variability in SWE during current years. We have also included three years of data to evaluate the accuracy of the model between years in the revised manuscript, but this is not intended to suggest anything about climate change.

Pg 4, Ln 7 & 19 – don't need 'Description of the' in either sub heading.

This change was made.

Pg4, In 12 – 'which' is grammatically correct after a comma rather than 'that'.

This change was made.

Pg 4, Ln 25-27 – following Winstral et al., (2002) and subsequent papers by Winstral et al., was this used to calculate redistribution of snow (especially above tree line) in drifts which are very important hydrological areas to get SWE correct in a watershed?

Yes, these methods were used to calculate the upwind contributing area to calculate redistribution of snow in drifts across the landscape. Although snow redistribution is not as important in the warm maritime low elevation snowpacks characteristic of the Pacific Northwest, than in drier higher elevation continental snowpacks, we still felt it important to include wind as a driver of spatial variability in SWE. We have included the following sentence to clarify your question in the revised manuscript, "Upwind contributing area data, which captures the variability in snow deposition as a result of wind redistribution for each cell throughout the watershed (Winstral et al., 2002), was calculated following Molotch et al., (2005)."

Pg 5, Ln 18-21 – while Sproles et al. (2013) is often cited, as this is such a key foundation to this work it needs greater explanation in this paper – in particular how the future SWE conditions are calculated, and especially the change to precipitation rates and phase (rain/snow) as well as temperature. A fairly detailed paragraph describing the methodology used for the modelled input SWE data was included in the data sources section of the methods. For the sake of brevity, we would prefer to cite Sproles et HESSD
al 2013, and not reiterate what has already been published. We include the following paragraph in the methods, "Modelled and gridded SWE data across the MRB (Figure 2) were provided by Sproles et al., (2013). These data were developed using a physicallybased spatially distributed snow mass and energy balance model, SnowModel (Liston and Elder, 2006). SnowModel uses micrometeorological and topographic data to distribute snow across the landscape accounting for climatic, topographic, and vegetation variability. The model was modified by Sproles et al., (2013) to account for rain/snow precipitation phase partitioning, and snow albedo decay in forested landscapes. This model was calibrated and validated using data from the four SNOTEL sites, meteorological data from the HJ Andrews Long Term Ecological Research site and National Weather Service stations and Landsat fractional snow covered area data over the sampling period 1989-2009 (Sproles et al., 2013). The model was run at 100-m spatial resolution on a daily time step. We used modelled peak SWE data as the predicted variable in the BRT model. Sproles et al., (2013) showed that 2009 was considered an average snow year so we used peak SWE from 2009 (five days centred on 04 April 2009) as our reference year. Additionally we used peak SWE from 2008 (five days centred on 24 April 2008) as an above average snow year, and peak SWE from 2005 (five days centred on 20 April 2005) as a below average snow year."

Pg 5, ln 24 – can more be said about issues of up-scaling (aggregation) and downscaling (disaggregation) of different data sets?

We have included additional information about the method used in scaling the input data following, "All spatial data were masked to the McKenzie River Basin and converted to the same projection and spatial resolution: NAD83, UTM Zone 10, and a 100-m grid cell size. Spatial data were processed using ArcGIS 10.1 using bilinear interpolation for continuous data and nearest neighbour interpolation for discrete data."

Pg5, Ln 25 – why concentrate on areas defined as 'bulk' rather than fully spatially distributed models? Locating big drifts, often above tree line, are key to understanding the timing and magnitude of discharge. This seems to have been neglected under this
BRT model.

The bulk snowpack was evaluated for current and future (+  $2^{\circ}$  C) conditions to quantify the area on the landscape where the majority of the snowpack lies from the physicallybased spatially-distributed modelled SWE data. Although we have removed this analysis from this revised manuscript to focus on the novel method of coupling physicallybased and statistical models to improve observational network design.

Pg 6, Ln 5-10 – The way that SnowModel is combined or used to evaluate BRT is presented in a very confusing fashion. Where is the independent data to evaluate BRT?

Within the CART statistical software, we have reserved 20,000 random cells within the modelling domain to test the final BRT statistical model. The modelled SWE data was used as the dependent variable in the BRT statistical model, and we hope this is now less confusing in the revised manuscript. We have included this additional information in the statement, "Within the CART software, the final BRT model was validated using reserved data from an independent set of 20,000 randomly selected grid cells from within the MRB."

Pg 6, In 12 - 20 BRT snow classes? Wasn't one removed due to logistics and finance? This adds confusion to the methods.

We have revised our analysis in this revised manuscript in a few key ways including, a) using peak SWE instead of 01 April, and b) pruning the optimal model to just the two main drivers to prevent overfitting and multi-collinearity brought up by reviewer #3. In the revised manuscript, the optimal model is defined by 21 BRT snow classes, none of which were removed. We hope this is clear in this revised manuscript. Pg 6, In 14-16 – Why were lower elevation extents removed? This is done without any quantification nor real justification.

The BRT model did not set bounds on lower elevation limits for SWE, although SWE did
not exist below approximately 600 m in the modelled data as well as anecdotally from observations. As stated in the manuscript, "Because the BRT-model did not determine a lower elevation limit on snow extent, we excluded areas with an elevation less than 600 m to prevent over-prediction of snow-covered area below elevations where it was observed in the modelled data."

Pg 6, In 16-17 – what proportion of the basin was removed? Why do this if it is a SWE contributing area to discharge, why would this cause over prediction?

This removed 26.5% of the area in the basin during the evaluation of the volume of SWE in the BRT classes, although this area held 0.068% of the SWE in the basin (or 0.0009 km3 of 1.48 km3 SWE). If we had not set a lowest extent of snow covered area in the basin, it would be the equivalent of drawing a regression line beyond the range of the input data. We set this boundary were it was observed in the modelled data to prevent extrapolation of the model beyond the bounds of the input data.

Pg 6, In 21 – add 'a' between 'create' and 'set'.

This change was made.

Pg 6, ln 24 – why is 500m threshold applied? In practice one would expect field locations for met sites to be closer or further away from transport links depending on local conditions (i.e. how potential met site locations have always previously been evaluated).

Your question highlighted a typo in the manuscript which we have revised, as well as included additional information for clarity. The following statement was included in the revised manuscript, "To prevent contamination from the road network, but still define accessible site locations, we also identified areas within 100-500 m of a snowmobile-accessible road."

Pg 7, Ln 8 – the 'final' BRT model. How many BRT models were evaluated? The rest of this paragraph has already been discussed and is providing repetition.

HESSD
In the revised manuscript, we developed the optimal model based on an average year snowpack, which was then paired down to a more parsimonious final optimal model, which was then applied to an above average and below average snowpack. In the revised manuscript, we have rewritten the methods for clarity, including the following text in the analysis section, "An optimal tree was produced to minimize the standard error of the model, which was then pruned down to the simplest tree possible within one standard error of the optimal tree, and so each terminal node represented at least 1% of the variability in peak SWE. The resultant tree identified 21 terminal nodes that characterized the spatial variability in SWE through combinations of independent drivers into 21 BRT-derived snow classes (Table 1). The BRT model identified elevation, land cover, NDVI, insolation, percent canopy cover, slope, and wind as significant explanatory drivers of the spatial variability of peak SWE (all selected variables had p-values

In this study we sought to classify the spatial distribution of snow water volume on the landscape, and therefore focused on the peak snow accumulation across the watershed. Aspect should matter, as well as insolation and slope, during snow ablation, but we did not expect it to be important in accumulation processes. Because we focused on snow accumulation, we did not expect aspect to be an important driver in the spatial distribution of peak SWE.

Pg 7, Ln 18 – why were BRT and SnowModel not used in conjunction with each other. When both are available it is confusing that they are not used together to optimize estimation of SWE distribution.

We coupled SnowModel, BRT, and a geospatial model to classify snow across the watershed. They are used in conjunction and hope we clarified this issue in the revised manuscript.

Pg 7, Ln 20 – BRT estimation of mass should be good in 2009 as it is tuned with SnowModel, but poor prediction of SCA (64% SCA over prediction) suggests it's not getting SWE right for the right spatial reasons (i.e. at low elevation).

We were previously using the static BRT-derived snow classes based on 2009 SWE and applying them to 2012. This method implies there is no inter-annual variability in SWE, and therefore does not properly evaluate the accuracy of the BRT model between years. In order to improve upon this, we included three years of SWE input data to develop three equivalent BRT models, and using omission and commission statistics we evaluated the overall accuracy of BRT models between years in the revised manuscript. We don't expect the BRT model to predict actual SWE volume on the landscape, but to predict the spatial distribution of similar SWE characteristics across the landscape, and we believe we have achieved this goal in this revised manuscript.

Pg 7, Ln 23 – Increasing elevation does not increase accumulation, it is increases with elevation (i.e. not a cause in itself).

HESSD
We have included the following statement to address this comment, "Snowpack accumulation increased with increasing elevation, resulting in a greater mean SWE per unit area at the highest elevations. Although deep snowpack at the highest elevations only cover a small aerial extent of the MRB, which resulted in decreasing contribution of total basin-wide SWE above approximately 1700 m during the average and above average snow years. In contrast, during the low snow year, the highest elevation classes contributed the most to total basin-wide SWE (Figure 5)."

Pg7, Ln 26-31 - could this information be put into a table?

This information has been refined and added to Table 1 in the revised manuscript.

Pg 8, Ln 1 – comma needed after 'Whereas'.

This change was made.

Pg 8, In 4-5 – How does BRT adapt to changes in winter precipitation inter-annually? If it can't, what advantages does it have over running SnowModel?

I hope this confusion has been clarified in this revised manuscript. We used the BRT model to classify the modelled SWE output of continuous data based on physiographic landscape characteristics. In order to address how the BRT model adapts to interannual variability in SWE, we included an above average snow year, and a below average snow year in the analysis of this revised manuscript.

Pg 8, Ln 7 – SnowModel derived estimates were NOT captured well by BRT. They were an order of magnitude different at low and medium elevations. Need a much better quantified argument to justify this.

We aim to capture the spatial variability in SWE characteristics across the landscape, not the actual volume of SWE. In the revised manuscript, we have rerun the BRT models using three years of data to more robustly evaluate the spatial variability in snow classes across the landscape between years.

HESSD
**Pg 8, Ln 13 – Need to provide more about how accessibility is determined as a criteria.**

These criteria have been described in the methods section including the following statement, "To create a set of feasible locations for the in situ snow monitoring network we evaluated the accessibility of locations within the MRB. Using a GIS-based binary selection model, we masked out all private lands and public lands where the presence of endangered Northern Spotted Owl prevented permitted access. To prevent contamination from the road network, but still define accessible site locations, we also identified areas within 100-500 m of a snowmobile-accessible road. From these accessible areas, the final sites were then randomly selected from each of the dominant BRT-derived snow classes within the seasonal snow zone." Also to clarify this in the results section we included the following statement, "The geospatial selection model identified 16 of the 21 classes as being accessible (following criteria explained in the above methods) during winter."

Pg 8, In 19 – six met stations is a bit misleading, rather there are three sites, each with adjacent open/forest met stations.

Although the six met stations are paired by elevation and appear to be in the same site on the map, they are approximately 1 km from each another. They are grouped by elevation but distinct in the land cover characteristics. The following statement has been included which clarifies that the six stations are grouped by three elevation ranges and two land cover types, "Within the area covered by these 16 classes, random site locations were selected within the six most abundant classes across the MRB to capture low, medium, and high elevations, with forested and open land cover classes. The resultant Forest Elevation Snow Transect (ForEST) monitoring network site locations were thus objectively selected to sample across the range of spatial variability in SWE. The ForEST network, composed of six meteorological stations and snow survey transects, was deployed in November 2011, and continues to provide high quality snow and climate data to evaluate snow-forest-climate interactions in the MRB (Figure 4)." HESSD
Pg 8, Ln 17-26 – this isn't a scientific result unless you then go on to do something with these met data.

The ForEST network of snow monitoring stations is the result of the coupled modeling approach, and therefore we believe it should be included in the results section. We have moved any qualitative evaluation of the network to the discussion section. We could fill an entire manuscript evaluating the met data, but in this manuscript we aim to focus on the novel method of objective site selection.

Pg 8, Ln 26 – how has this been stringently validated with the BRT model?

We aimed to objectively distribute a representative snow monitoring network using this coupled modeling approach. Instead of heuristically deciding where stations should be located, we used physically based SWE data and a non-parametric statistical model to define the spatial variability across the watershed. We hope in the revised manuscript we have clarified this and include the following statement in the discussion section, "The paired forest-open land cover site selection process has already led to important understanding of key sub-canopy snow processes (Storck et al., 2002; Golding and Swanson, 1986). But here, the assumptions driving paired site selection process has been further validated using coupled physically-based spatially-distributed snow model input data and non-parametric BRT statistical modelling across a forested montane watershed."

Pg 8, Ln 26-28 – Consistency in the pattern of measured snow course SWE doesn't corroborate energy balance and snow-veg interactions.

We have removed this statement to allow the reader to form his/her own conclusions about the measured data resulting from this project.

Pg 9, Ln 15-16 – This study doesn't explicitly demonstrate the impact of timber harvest/ fire disturbance impact on SWE distribution.

We have included the following statement in the discussion to clarify this point, "By
distinguishing snow classes based on forest vs. open land cover across a range of elevations, this study emphasizes the watershed-scale control that vegetation and particularly land cover change relative to timber harvest (and potentially fire disturbance) has on snowpack accumulation in the maritime western Oregon Cascades."

Pg 9, Ln 20-21 – If BRT and SnowModel are coupled (as stated) then what does this combination give us that SnowModel doesn't give us as a stand-alone product? This is not providing added information on hydrological response units (HRU), it is not a new idea in snow hydrology (e.g. CRHM), and doesn't provide an obvious robust advancement in inter-annual transferability.

We use this coupled approach to classify snow characteristics across the landscape and to improve upon traditional methods of observational network design. The Snowmodel output data is continuous by nature, and doesn't provide any guidance for which specific locations in a watershed may be representative of greater landscape scale processes. We did not expect to advance scientific knowledge, but to provide an objective technique for distributing point based monitoring locations which represent the spatial variability across the watershed.

Pg 9, Ln 26 – Yes, inter-annually transferability really needs to be more robustly tested by this methodology, rather than one 1 April snapshot in 2012. Currently this evaluation/validation has not been sufficiently done with independent data.

We included three years of data from the actual date of peak SWE in this revised manuscript to provide a more robust evaluation of the accuracy of the BRT models between years.

Table 1 – What percentage of SCA was above 1546m (was it 40%)? If these data were rejected can this be demonstrated that this is not a problem? While thin SWE and scour is likely in Alpine areas above tree line drifts in these areas can contribute substantially to the timing of increased discharge through melt-out.
We aim to capture the spatial variability in SWE characteristics across the watershed to improve observational network design, not to accurately capture any watershed discharge characteristics. We do need more observation stations at higher elevations, although within the resource limitations of this study we were restricted to locations we could reach with a snowmobile and must accept the related uncertainty.

Table 2 – no units. Can low, medium and high be classified? Which sites were the forest and open sites – can these be related to a map or specifically described?

Table 2 has been omitted from this revised manuscript in lieu of more robust accuracy assessment tables included in the supplementary tables.

Fig 2 – put yellow circles in legend. Cite Sproles in caption (see previous comment about more explicit explanation of future precipitation scenario in Sproles data).

Figure 2 has been omitted from the revised manuscript as described in the above text.

Fig 3 – I am surprised that mean SWE by elevation increased above tree-line, would have expected some thinning of SWE due to scour, can this be explained? The hypsometry of the basin would be a very useful (essential?) addition to this figure.

As mentioned above, snow redistribution is not as important in warm maritime snowpacks as it is in cold dry continental snowpacks, and therefore we are not surprised by this result. In order to make the connection between volumetric SWE and mean SWE across the elevation gradient we have included the hypsometry on Figure 3.

Fig 4 – relate snow classes to the Table otherwise they make no sense.

Table 1 has been altered in the revised manuscript to explicitly describe each snow class for each year.

The BRT snow class numbers in Table 1 match those used in the legend of Figure 4 in order for readers to make the connection between the statistics and spatial distribution of each snow class.

HESSD
Fig 5 – How does forest and open relate to the 'all' classification? What is additional to 'all' other than forest and open? Why is mean SWE so different to SnowModel? Which year is this for? Don't put descriptive results in caption, put them in the main body of the text. Caption says it's statistically important, where is this statistical analysis?

The information for each BRT-derived snow class is now consistent between Table 1, Figure 4 and Figure 5 in the revised manuscript. The BRT model distinguishes snow classes across the middle elevations into forest vs. open land cover types, but only by elevation across the high elevations. All land covers includes both forest and open land covers as opposed to forest or open land covers. Mean SWE here is defined for each BRT snow class, where as in Figure 3 mean SWE is defined for each 100 m elevation band. We hope these questions are addressed in the revised manuscript. The BRT model selected statistically significant drivers of SWE across the landscape and is the analysis we refer to here, although these descriptive results have been removed from the figure caption.

Fig 6 – This is just measured SWE, how is it use to quantitatively evaluate the new modelling framework? Need to define the high, mid and low elevations in the caption. Error bars seem to be the range rather than any calculation of error.

We are not evaluating the modeling framework using these measured SWE data, but we present the measured SWE data to show there are consistent differences in peak SWE between forests and open areas that seem to evolve across the elevational gradient. We have included the elevations within the caption, and also state in the caption that, "Error bars indicate the maximum and minimum measured SWE (cm) from 2012, 2013, and 2014." We are unaware of another name to refer to these bars and hope they are clearly defined in the revised manuscript.

Thank you very much for your considerate review of our manuscript.

Please also note the supplement to this comment:
http://www.hydrol-earth-syst-sci-discuss.net/hess-2016-317/hess-2016-317-AC1-supplement.pdf

**HESSD**
**HESSD**
Table S1. Accuracy assessment matrix comparing the BRT classes derived from the normal snow year 2009 with those from the high snow year 2008. Overall there is less error in the lowest and highest elevation BRT classes, whereas the mid-elevations there is more error between models. Many classes were reassigned when the BRT model was rerun between years, underestimating the accuracy of the overall spatial variability between models.

| 2009   | 1     | 2     | 3     | 4    | 5    | 6    | 7    | 8    | 9    | 10   | 11   | 12   | 13   | 14    | 15   | 16    | 17    | 18    | 19    | 20      | 21    | Comission |
|--------|-------|-------|-------|------|------|------|------|------|------|------|------|------|------|-------|------|-------|-------|-------|-------|---------|-------|-----------|
| 2008   |       |       |       |      |      |      |      |      |      |      |      |      |      |       |      |       |       |       |       |         |       | error (%) |
| 1      | 55402 | 6035  |       |      |      |      |      |      |      |      |      |      |      |       |      |       |       |       |       |         |       | 10        |
| 2      |       | 16467 |       |      |      |      |      |      |      |      |      |      |      |       |      |       |       |       |       |         |       |           |
| 3      |       | 369   | 22960 |      |      |      |      |      |      |      |      |      |      |       |      |       |       |       |       |         |       |           |
| 4      |       | 52    |       | 3930 |      |      |      |      |      |      |      |      |      |       |      |       |       |       |       |         |       |           |
| 5      |       |       |       |      | 9879 |      |      |      |      |      |      |      |      |       |      |       |       |       |       |         |       | (         |
| 6      |       |       |       |      | 5486 |      |      |      |      |      |      |      |      |       |      |       |       |       |       |         |       | 10        |
| 7      |       |       |       |      |      | 3232 | 3232 |      |      |      |      |      |      |       |      |       |       |       |       |         |       | 5         |
| 8      |       |       |       |      |      |      |      | 4667 |      |      |      |      |      |       |      |       |       |       |       |         |       | (         |
| 9      |       |       |       |      |      |      |      |      | 2524 |      |      |      |      |       |      |       |       |       |       |         |       |           |
| 10     |       |       |       |      |      |      |      | 2053 |      | 4007 |      |      |      |       |      |       |       |       |       |         |       | 3         |
| 11     |       |       |       |      |      |      |      |      |      | 5276 | 5740 |      |      |       |      |       |       |       |       |         |       | 4         |
| 12     |       |       |       |      |      |      |      |      | 486  |      |      | 2900 |      |       |      |       |       |       |       |         |       | 1         |
| 13     |       |       |       |      |      |      |      |      |      |      | 1965 | 339  | 5421 |       |      |       |       |       |       |         |       | 3         |
| 14     |       |       |       |      |      |      |      |      |      |      |      |      | 5252 | 4338  | 617  |       |       |       |       |         |       | 5         |
| 15     |       |       |       |      |      |      |      |      |      |      |      |      |      | 13692 | 1948 | 719   |       |       |       |         |       | 8         |
| 16     |       |       |       |      |      |      |      |      |      |      |      |      |      |       |      | 10260 | 14155 |       |       |         |       | 5         |
| 17     |       |       |       |      |      |      |      |      |      |      |      |      |      |       |      |       |       | 23580 |       |         |       | 10        |
| 18     |       |       |       |      |      |      |      |      |      |      |      |      |      |       |      |       |       |       | 5931  | 705     |       | 10        |
| 19     |       |       |       |      |      |      |      |      |      |      |      |      |      |       |      |       |       |       |       | 1850    |       | 10        |
| 20     |       |       |       |      |      |      |      |      |      |      |      |      |      |       |      |       |       |       |       | 1057    | 1025  | 5         |
| 21     |       |       |       |      |      |      |      |      |      |      |      |      |      |       |      |       |       |       |       |         | 2039  |           |
| ission | 0     | 28    | 0     | 0    | 36   | 100  | 0    | 31   | 16   | 57   | 26   | 10   | 49   | 76    | 24   | 7     | 100   | 100   | 100   | 71      | 33    |           |
| or (%) |       |       |       |      |      |      |      |      |      |      |      |      |      |       |      |       |       |       | Overa | II accu | iracy | 63        |

Fig. 1. Supplemental Table 1\_Accuracy Accessment

**HESSD**
Table S2. Accuracy assessment matrix comparing the BRT classes derived from the normal snow year 2005 with those from the high snow year 2008. Overall there is less error in the lowest and highest elevation BRT classes, whereas the mid-elevations there is more error between models. Many classes were reassigned when the BRT model was rerun between years, underestimating the accuracy of the overall spatial variability between models.

| 2009   | 1     | 2     | 3     | 4    | 5     | 6    | 7    | 8    | 9    | 10   | 11   | 12   | 13    | 14    | 15   | 16   | 17   | 18   | 19   | 20   | 21   | Comission |
|--------|-------|-------|-------|------|-------|------|------|------|------|------|------|------|-------|-------|------|------|------|------|------|------|------|-----------|
| 2005   |       |       |       |      |       |      |      |      |      |      |      |      |       |       |      |      |      |      |      |      |      | error (%) |
| 1      | 55402 | 22923 | 22960 | 3930 | 15365 | 3232 | 6013 | 3365 | 2243 |      |      |      |       |       |      |      |      |      |      |      |      | 5         |
| 2      |       |       |       |      |       |      |      | 3355 |      | 9283 | 5840 |      |       |       |      |      |      |      |      |      |      | 10        |
| 3      |       |       |       |      |       |      |      |      | 767  |      |      | 2900 |       |       |      |      |      |      |      |      |      | 10        |
| 4      |       |       |       |      |       |      |      |      |      |      | 1965 |      | 9212  | 12939 |      |      |      |      |      |      |      | 10        |
| 5      |       |       |       |      |       |      |      |      |      |      |      |      |       | 5091  | 757  | 3973 |      |      |      |      |      | 10        |
| 6      |       |       |       |      |       |      |      |      |      |      |      | 339  | 1461  |       | 1808 | 879  |      |      |      |      |      | 10        |
| 7      |       |       |       |      |       |      |      |      |      |      |      |      |       |       |      | 3718 |      |      |      |      |      | 10        |
| 8      |       |       |       |      |       |      |      |      |      |      |      |      |       |       |      |      | 2194 |      |      |      |      | 10        |
| 9      |       |       |       |      |       |      |      |      |      |      |      |      |       |       |      |      | 3622 |      |      |      |      | 10        |
| 10     |       |       |       |      |       |      |      |      |      |      |      |      |       |       |      |      | 2697 |      |      |      |      | 10        |
| 11     |       |       |       |      |       |      |      |      |      |      |      |      |       |       |      |      | 3702 |      |      |      |      | 10        |
| 12     |       |       |       |      |       |      |      |      |      |      |      |      |       |       |      |      | 1815 |      |      |      |      | 10        |
| 13     |       |       |       |      |       |      |      |      |      |      |      |      |       |       |      |      |      | 7239 |      |      |      | 10        |
| 14     |       |       |       |      |       |      |      |      |      |      |      |      |       |       |      |      |      | 4776 |      |      |      | 10        |
| 15     |       |       |       |      |       |      |      |      |      |      |      |      |       |       |      |      |      | 4045 |      |      |      | 10        |
| 16     |       |       |       |      |       |      |      |      |      |      |      |      |       |       |      |      |      | 2347 |      |      |      | 10        |
| 17     |       |       |       |      |       |      |      |      |      |      |      |      |       |       |      |      |      | 3253 |      |      |      | 10        |
| 18     |       |       |       |      |       |      |      |      |      |      |      |      |       |       |      |      |      | 1923 | 512  |      |      | 2         |
| 19     |       |       |       |      |       |      |      |      |      |      |      |      |       |       |      |      |      |      | 3857 |      |      |           |
| 20     |       |       |       |      |       |      |      |      |      |      |      |      |       |       |      |      |      |      | 1562 | 3612 | 421  | 3         |
| 21     |       |       |       |      |       |      |      |      |      |      |      |      |       |       |      |      |      |      |      |      | 2643 |           |
| ission | 0     | 100   | 100   | 100  | 100   | 100  | 100  | 100  | 100  | 100  | 100  | 100  | 100   | 100   | 100  | 100  | 100  | 92   | 35   | 0    | 14   |           |
| or (%) |       |       |       |      |       |      |      |      |      |      |      |      | uracv |       | 2    |      |      |      |      |      |      |           |

Fig. 2. Supplemental Table 2\_Accuracy Accessment

---

## Author Comment (AC2) · 5 Nov 2016

Dear Reviewer,

Thank you for your comments and recommendations for the revised manuscript, they were very helpful in presenting this research in a more robust and defensible way. In order to "close the circle" and tell a more compelling story, we have made multiple changes to the revised manuscript. We focused the paper solely on the objective approach to improve snow observational network design, and therefore omitted the evaluation of the SNOTEL network under climate change. We acknowledge the limitation in the initial analysis conducted in 2010 which was based on data from 01 April 2009, with the assumption it represented maximum snow accumulation across the basin during an

average snow year. To improve upon this in the revised manuscript we used data from the five days centered on the date of actual peak SWE in the McKenzie River Basin for an average year 2009, an above average year 2008, and a below average year 2005. Evaluating the BRT-derived snow classes from three years of SWE data enabled us to use a more robust analytical approach including omission and commission statistics of overall classification accuracy.
Gleason et al. present a case study detailing how a binary regression tree (BRT) model can be used to identify major statistical classes of snow accumulation based on readily observable physiography. This regression tree classification/ model subsequently is used to inform a detailed snowpack monitoring array/ network. The specific application is in the MacKenzie River basin in the Cascade Mountains of Oregon, an area subject to strong elevation gradients in snowpack accumulation and potential quite susceptible to climate change. The result of this effort was the establishment of the Forest Elevational Snow Transect (ForEST) which has been operational for five years. The case is made that an objective approach like the BRT is preferable to errors associated with a subjective siting based on heuristics or experience. Specifically, the drivers are: elevation, land cover, percent canopy cover, slope, NDVI, and latitude. Not terribly surprising, but nice to see objectively defined.

Although technically sound, the paper could do a better job of closing the circle on a compelling story or novel contribution. Discussion of climate change, monitoring, and site selection are mixed making it difficult to determine exactly what is the contribution of this effort beyond adding another site at slightly higher elevation. A clearer analysis of how the BRT guided siting resulting in a more representative/ predictive/ useful network in a climate change scenario is needed. For example, how can others use this approach without physically validated SWE distributions? Should we expect this classification to hold elsewhere or even here under climate change? Indeed, should we move the discussion beyond April 1st to better inform water resource management? That date is a compromise after all and it very well may be that a more representative/ useful for water resources monitoring network is not wedded to that date (heresy I know). I can't say which of the above (or others) are most worthwhile avenues to pursue, but after reading the manuscript several times, I am left with the sense that something is missing.

Specific Comments

Although there is a certain level of objectivity in the BRT, I suggest that the modeling approach carries its own set of biases first in the physical model which must scale point observations to a 3-D snow cover while only the wind redistribution part of the model operates in 3-D including non-local effects. A number of papers suggest that edge effects of vegetation on energy balance as well as remote topographical vs local vegetation shading influence snowpack mass and energy balance. These are difficult to include, especially at the scale of this exercise, but they clearly may be relevant for future scenarios. Instead, the strength (of using the BRT and physical model) is being able to accurately evaluate these assumptions by making them explicit, rather than suggesting they don't exist. No model is perfect.

In the discussion we acknowledge the many potential sources of uncertainty in this method, in the following statement, "This objective method is a useful tool in classifying snow characteristics across the landscape to determine representative locations for intelligent snowpack monitoring particularly in physiographically complex landscapes. Although it is an improvement over more commonly used heuristic approaches to site selection, the method incorporates uncertainty as a result of compounding statistically-, physically-, and spatially-based models which justifies caution in implementing these estimates in management decisions. However, the method meets assumptions of nonparametric data analysis, is performed with relative ease, and if data are available for the research basin of interest, it can be well validated. As even physically-based models incorporate inherent empirically-based historically-derived assumptions, there is also uncertainty in using this approach to represent future spatial variability in snow accumulation."

Second, the statistical approach used in the BRT assumes stationairity in the processes from year to year for your comparisons as well as for future predictions under climate change. This leaves me confused by the 50% difference in BRT vs physical model SWE for 2012. How was the BRT initialized for 2012? Did you force the tree structure to replicate that from 2009 or did you let the model form its own structure?

Yes, good point. We reran the BRT models using peak SWE for three years of data to develop a more robust comparison of the spatial distribution in BRT-derived snow classes between years. In the revised manuscript we used the average year SWE data to build an optimal BRT model, then used the same parameter conditions with SWE data from an above average year and below average year and let the model form its own structure. We then used omission and commission statistics to compare the accuracy in the spatial distribution of these BRT-derived snow classes between years.

The observation that patterns persist is nice, but it seems that there is information in the differences between the two estimates of SWE volume. I suggest showing the tree and order/ strength of nodes. This would strengthen the presentation of the BRT relative to the table.

In the revised manuscript, we simplified the final BRT model to just two predictive variables to reduce multi-collinearity between variables and prevent overfitting the final model. It is a good suggestion to show the BRT tree for clarity, however now that the final BRT model has been simplified, we think it is redundant to show the associated tree.

Figure 3 – At first glance (and second) it seems that only one of the new locations is

in the area of volumetric SWE accumulation. In other words, one location is too low in elevation and there is a large volume higher in elevation that your highest site. Just eyeballing it perhaps 30 to 40% of SWE is above that location.

To address comments on the stationarity assumptions and to simplify the overall manuscript we have excluded the discussion of climate change throughout the revised manuscript. We acknowledge that the ForEST station locations are not distributed in space, but span the parameter space in the spatial variability of SWE across the MRB for an average year, above average, and below average year. There is a large area of SWE above our highest site which was not feasible for us to monitor.

Figure 5 - I find this confusing - the classes open, forest, and all don't seem to make sense. perhaps you are trying to communicate too many messages? area, volume, and controls all in one figure with no indication that the volume differences between forest and open are due to area or depth

We have modified this figure in the revised manuscript, and hope it is easier to understand. Also we included a discussion of how interrelated the volume differences between forest and open areas are a function as both area and depth. The following has been included in the revised manuscript in the caption for Figure 5, "In the mid-elevations, the forest vs. open distinction is statistically important in distinguishing snow classes. In the high-elevations, above treeline, only elevation significantly drives variability in snow accumulation. Mean SWE increases but volumetric SWE decreases as the land area decreases at the highest elevations."

Figure 6 why show max and minimum as it muddies the difference between location and year. We think it is important to show the variability as well as the average snow accumulation between sites and across years.

Technical comments

Page 1, Lines 22-26 – It seems that the current ForEST network of sites is the result of

the BRT modeling exercise, if so, say that directly rather than back in to it as currently written We included the following statement to address this comment, "The Forest Elevational Snow Transect (ForEST) is a result of the BRT modelling and represents combinations of forested and open land cover types at low, mid, and high elevations."

Page 4, Line 25 and on to next page - Unclear how NLCD, 30m LANDFIRE, and 250m NDVI were used (what data from each source) and/ or aggregated to reach 100m spatial resolution. Specify this up front so the reader doesn't need to go back and forth between results and methods

We included the following statement in the methods section to specify how the data were aggregated, "All spatial data were masked to the McKenzie River Basin and converted to the same projection and spatial resolution: NAD83, UTM Zone 10, and a 100-m grid cell size. Spatial data were processed using ArcGIS 10.1 using bilinear interpolation for continuous data and nearest neighbour interpolation for discrete data."

Land cover is variously referred to as forest or open, and is synonymous with veg class, correct? This should be clarified.

Land cover is used throughout the revised manuscript, and veg class has been omitted. Table 2 – what are units?

Table 2 has been removed from the revised manuscript.

Thank you very much for our considerate review of our manuscript.

Sincerely, Kelly Gleason

Please also note the supplement to this comment:
http://www.hydrol-earth-syst-sci-discuss.net/hess-2016-317/hess-2016-317-AC2-supplement.pdf

[Figure]

Table S1. Accuracy assessment matrix comparing the BRT classes derived from the normal snow year 2009 with those from the high snow year 2008. Overall there is less error in the lowest and highest elevation BRT classes, whereas the mid- elevations there is more error between models. Many classes were reassigned when the BRT model was rerun between years, underestimating the accuracy of the overall spatial variability between models.

| BRT Class 2009 / 2008 | 1 | 2 | 3 | 4 | 5 | 6 | 7 | 8 | 9 | 10 | 11 | 12 | 13 | 14 | 15 | 16 | 17 | 18 | 19 | 20 | 21 | Comission error (%) |
|---|---|---|---|---|---|---|---|---|---|---|---|---|---|---|---|---|---|---|---|---|---|---|
| 1 | 55402 | 6035 | | | | | | | | | | | | | | | | | | | | 10 |
| 2 | | 16467 | | | | | | | | | | | | | | | | | | | | 0 |
| 3 | | 369 | 22960 | | | | | | | | | | | | | | | | | | | 2 |
| 4 | | 52 | | 3930 | | | | | | | | | | | | | | | | | | 1 |
| 5 | | | | | 9879 | | | | | | | | | | | | | | | | | 0 |
| 6 | | | | | 5486 | | | | | | | | | | | | | | | | | 100 |
| 7 | | | | | | 3232 | 3232 | | | | | | | | | | | | | | | 50 |
| 8 | | | | | | | | 4667 | | | | | | | | | | | | | | 0 |
| 9 | | | | | | | | | 2524 | | | | | | | | | | | | | 0 |
| 10 | | | | | | | | 2053 | | 4007 | | | | | | | | | | | | 34 |
| 11 | | | | | | | | | | 5276 | 5740 | | | | | | | | | | | 48 |
| 12 | | | | | | | | | 486 | | | 2900 | | | | | | | | | | 14 |
| 13 | | | | | | | | | | | 1965 | 339 | 5421 | | | | | | | | | 30 |
| 14 | | | | | | | | | | | | | 5252 | 4338 | 617 | | | | | | | 57 |
| 15 | | | | | | | | | | | | | | 13692 | 1948 | 719 | | | | | | 88 |
| 16 | | | | | | | | | | | | | | | | 10260 | 14155 | | | | | 58 |
| 17 | | | | | | | | | | | | | | | | | | 23580 | | | | 100 |
| 18 | | | | | | | | | | | | | | | | | | | 5931 | 705 | | 100 |
| 19 | | | | | | | | | | | | | | | | | | | 1850 | | | 100 |
| 20 | | | | | | | | | | | | | | | | | | | 1057 | 1025 | | 51 |
| 21 | | | | | | | | | | | | | | | | | | | | | 2039 | 0 |
| Omission error (%) | 0 | 28 | 0 | 0 | 36 | 100 | 0 | 31 | 16 | 57 | 26 | 10 | 49 | 76 | 24 | 7 | 100 | 100 | 100 | 71 | 33 | |
| | | | | | | | | | | | | | | | | | | | Overall accuracy | | | 63 |

**Fig. 1.** Supplementary Table 1_Accuracy Accessment

Table S2. Accuracy assessment matrix comparing the BRT classes derived from the normal snow year 2005 with those from the high snow year 2008. Overall there is less error in the lowest and highest elevation BRT classes, whereas the mid- elevations there is more error between models. Many classes were reassigned when the BRT model was rerun between years, underestimating the accuracy of the overall spatial variability between models.

| BRT Class 2005 \ 2009 | 1 | 2 | 3 | 4 | 5 | 6 | 7 | 8 | 9 | 10 | 11 | 12 | 13 | 14 | 15 | 16 | 17 | 18 | 19 | 20 | 21 | Comission error (%) |
|---|---|---|---|---|---|---|---|---|---|---|---|---|---|---|---|---|---|---|---|---|---|---|
| 1 | 55402 | 22923 | 22960 | 3930 | 15365 | 3232 | 6013 | 3365 | 2243 | | | | | | | | | | | | | 59 |
| 2 | | | | | | | | 3355 | | 9283 | 5840 | | | | | | | | | | | 100 |
| 3 | | | | | | | 767 | | | | | 2900 | | | | | | | | | | 100 |
| 4 | | | | | | | | | | 1965 | | | 9212 | 12939 | | | | | | | | 100 |
| 5 | | | | | | | | | | | | | | 5091 | 757 | 3973 | | | | | | 100 |
| 6 | | | | | | | | | | | 339 | 1461 | | | 1808 | 879 | | | | | | 100 |
| 7 | | | | | | | | | | | | | | | | 3718 | | | | | | 100 |
| 8 | | | | | | | | | | | | | | | | | 2194 | | | | | 100 |
| 9 | | | | | | | | | | | | | | | | | 3622 | | | | | 100 |
| 10 | | | | | | | | | | | | | | | | | 2697 | | | | | 100 |
| 11 | | | | | | | | | | | | | | | | | 3702 | | | | | 100 |
| 12 | | | | | | | | | | | | | | | | | 1815 | | | | | 100 |
| 13 | | | | | | | | | | | | | | | | | | 7239 | | | | 100 |
| 14 | | | | | | | | | | | | | | | | | | 4776 | | | | 100 |
| 15 | | | | | | | | | | | | | | | | | | 4045 | | | | 100 |
| 16 | | | | | | | | | | | | | | | | | | 2347 | | | | 100 |
| 17 | | | | | | | | | | | | | | | | | | 3253 | | | | 100 |
| 18 | | | | | | | | | | | | | | | | | 1923 | 512 | | | | 21 |
| 19 | | | | | | | | | | | | | | | | | | | 3857 | | | 0 |
| 20 | | | | | | | | | | | | | | | | | | 1562 | 3612 | 421 | | 35 |
| 21 | | | | | | | | | | | | | | | | | | | | | 2643 | 0 |
| Omission error (%) | 0 | 100 | 100 | 100 | 100 | 100 | 100 | 100 | 100 | 100 | 100 | 100 | 100 | 100 | 100 | 100 | 100 | 92 | 35 | 0 | 14 | Overall accuracy 28 |

**Fig. 2.** Supplementary Table 2_Accuracy Accessment

**Supplement:**

[revised manuscript text omitted]

---

## Author Comment (AC3) · 5 Nov 2016

Dear Reviewer,

Thank you for your comments and recommendations for the revised manuscript, they were very helpful in presenting this research in a more robust and defensible way. In order to tell a more compelling story, we have made multiple changes to the revised manuscript. We focused the paper solely on the objective approach to improve snow observational network design, and therefore omitted the evaluation of the SNOTEL network under climate change. We acknowledge the limitation in the initial analysis conducted in 2010 which was based on data from 01 April 2009, with the assumption it represented maximum snow accumulation across the basin during an average snow

year. To improve upon this in the revised manuscript we used data from the five days centered on the date of actual peak SWE in the McKenzie River Basin for an average year 2009, an above average year 2008, and a below average year 2005. Evaluating the BRT-derived snow classes from three years of SWE data enabled us to use a more robust analytical approach including omission and commission statistics of overall classification accuracy.
This paper (1) investigates which physiographic factors influence modeled spatial SWE distributions on 1 April in the McKenzie River Basin of the western Oregon Cascades and (2) demonstrates how this knowledge can be used to locate new snow studies sites in an objective way for resolving physiographic influences on SWE. The work is motivated to inform observational network design in snow-dominated watersheds where forest change and climate change present challenges. They use binary regression trees (BRT) to predict 1 April SWE in an average year based on predictors such as elevation, forest cover, NDVI, and latitude. This is a unique application of BRT because they use 1 April SWE output from a spatially distributed physically-based model (Snow-Model) at the watershed scale, whereas most previous BRT snow studies have been at smaller scales and with observational data. The analysis examines 20 snow classes from BRT in average snow years in current and future (+2 C degree) conditions. The study compares differences in SWE in forest and clearings at different elevations, as sampled in the ForREST network. I think the most major contribution of the paper is that it demonstrates a method for utilizing physically-based model output to improve observational network design. The method is novel and the results should garner decent interest from the community. I think the writing/figures are of especially high quality. This paper should be published in HESS after addressing a variety of major and minor

comments (below).

MAJOR SPECIFIC COMMENTS

1. The most glaring weakness of the analysis is that it does not address collinearity of the predictors anywhere. Certainly some of the predictors in the BRT co-evolve in space. For example, forest cover decreases with elevation (seen in Figure 3). How can one disentangle the unique influence of covarying predictors within the adopted regression framework, let alone assert which predictors dominate SWE (Page 7, Line 12)?

In order to reduce the multi-collinearity of the predictors and prevent overfitting the model we excluded predictors which explained less than 1% of the variability in SWE to develop a more parsimonious model. Elevation and land cover type explained 93% of the variability in SWE, which justified excluding the less important more correlated variables. We have confidence in this final model as it captures different patterns of influence from elevation and land cover to the spatial variability in snow accumulation between years. Across the basin, SWE increases predictably with elevation, but in the most forested regions of the river basin (mid-elevations) land cover also drives variability in SWE. Open areas tend to accumulate more SWE than forested areas at the same elevation, and forested areas tend to have more variability in SWE accumulation than open areas across elevation gradients. We have included the following statement to address this comment, "The BRT model identified elevation, land cover, NDVI, insolation, percent canopy cover, slope, and wind as significant explanatory drivers of the spatial variability of peak SWE (all selected variables had p-values < 0.05 and are listed above in order of significance). Although elevation and land cover were the dominant predictive variables where the other physiographic variables each explained less than 1% of the variability in peak SWE. In order to reduce the multi-collinearity between related variables and reduce the risk of overfitting the model, we simplified the final model to only include elevation and land cover. The final BRT model was validated using data for an independent set of 20,000 randomly selected grid cells from within the

MRB. The final parameters developed in this optimal tree for peak SWE in an average year 2009, were used to develop equivalent BRT models using peak SWE input for an above average year 2008, as well as to peak SWE during a below average year 2005."

2. The analysis implies that current observational sites may not be representative of snow conditions in a future climate and that physically-based model outputs are valid irrespective of climate conditions. However, SnowModel (and other models with the physically-based label) do have embedded routines/parameterizations that are empirical in nature and tuned to historical conditions (e.g., atmospheric longwave radiation). There is a general lack of discussion about the reliability of models in projecting changes outside of historical conditions. These approximations of the real-world are further muddied here because the study is advancing a model (BRT) of a model (Liston/ Elder SnowModel).

To focus the paper, we have removed the evaluation of SNOTEL sites under future climate change, and therefore removed the implication you mention "that physically-based model outputs are valid irrespective of climate conditions". We also included the following in the discussion at page 9, line 22, "As physically-based models incorporate inherent empirically-based historically-derived assumptions, there is also uncertainty in using this approach to represent future spatial variability in snow accumulation."

3. Comparing basin SWE on 1 April for the current climate and a warmer +2 degree climate may be misleading/inappropriate, as the basin may be well into the melt season by 1 April in the warmer climate. 1 April is historically significant only because it has been (on a mean basis) near peak SWE timing. Arguably, the date of peak SWE will advance earlier in the year with climate warming. So analyzing 1 April in a future warmer climate is like analyzing a date in mid- or late- April in the current climate, and we might say that SNOTEL sites are unrepresentative of basin conditions once melt conditions have advanced to that date in late April. However, that is not a fair comparison, as the SNOTEL sites may have been more representative of mean conditions earlier in the season (i.e., near peak conditions). To address this potential issue, the

authors should consider not only the spatial distribution of SWE but also the temporal evolution. Are the SNOTEL sites more representative of basin SWE at an earlier date (e.g., March 15) in the warmer climate?

We have removed this analysis in the final manuscript as discussed above in the general comments.

MINOR SPECIFIC COMMENTS

1. The "Future year (1 April 2012)" terminology versus +2 degree C year terminology is inconsistent and confusing at times. How can April 2012 be "a future year" when it is now (in 2016) well in the past (e.g., Page 6, Lines 19-20)? This needs better explanation. Also, please consider revising the language throughout the manuscript.

We have removed this analysis in the final manuscript as discussed above in the general comments. 2. The "high inter-annual variability in SWE" is offered as a reason for differences in SWE volume from BRT vs. SnowModel in the future scenario (page 8, line 4). However, this does not make sense, given that only average years are considered in the analysis, effectively precluding any influences of inter-annual variability. The authors go on to contradict the above assertion about inter-annual variability in the discussion: "This method could be improved by including more years of input data to fully capture the inter-annual temporal variability in the spatial distribution of SWE." Please revise.

To make the analysis more robust in this revised manuscript we included additional years of data for an above average year and a below average year to run the BRT model, in addition to the average snow year data we used in the first analysis. This has enabled us to use more robust quantitative evaluation of model accuracy between years using omission and commission statistics. We included the following describing these results, "The final optimal BRT model from the normal snow year (2009) applied to the high snow year (2008) demonstrated an overall accuracy of 63%, whereas the BRT model applied to the low snow year (2005) demonstrated an overall accuracy of

26% (Table S1, S2). The BRT model performed well across the low and high elevations, where errors of omission and commission were generally lowest (Table S1, S2). Although across the mid-elevations which consist of a patchwork of forest harvest and fire disturbance, were the areas with the greatest error between the BRT models. The high elevations above tree line, were the most consistently classified areas with low error between BRT models. The high error across the mid-elevations was due at least in part to the renumbering of classes when the model is rerun for each year, and therefore these statistics may underrepresent the accuracy of the BRT-model in predicting overall spatial patterns of physiographically derived snow classes between years. The BRT-modelled snow classes captured the spatial variability in SWE across the MRB relative to elevation and land cover during an average, above average, and below average snow year and were used to objectively inform the site selection of a snow monitoring network."

3. Was this analysis actually conducted prior to the installation of the ForEST network in November 2011? Or is this a retrospective analysis to test the representativeness of the established network? The connection between the presented work and the design of the ForEST network is never really made clear.

This distinction has implications for the title and tone of the manuscript. Currently, the manuscript implies that the analysis was used to inform the design of the ForEST network (page 10, lines 5-7). The current title is appropriate if the analysis with April 2009 was conducted first. However, if this is a retrospective analysis of the adequacy of the network, then the title may be better stated as "Testing the representativeness of a snow monitoring network in a forested mountain watershed".

Yes, this analysis was conducted prior to the installation of the ForEST network in 2010. Now I realize the need to emphasize the main narrative in this research, presenting the integrated objective method for informing snow observation network design. In the introduction and throughout the paper we have emphasized the connection between the presented work and the design of the ForEST network. We have augmented this

original analysis in the revised manuscript by including two additional years of data, but this does not alter the conclusions from the original analysis.

TECHNICAL CORRECTIONS

- Page 2, Line 28: Add "currently" before manages (the number of SNOTEL stations changes in time). Sentence changed to, "The NRCS currently manages approximately 858 Snowpack Telemetry (SNOTEL) stations across the western US (http://www.wcc.nrcs.usda.gov/snotel/SNOTEL_brochure.pdf)."

- Page 4, Line 9: "In the heart of" is somewhat colloquial; consider rephrasing this sentence.

Sentence changed to, "The McKenzie River, located in the western Oregon Cascades, is a major tributary of the Willamette River (Figure 1)."

- Page 4, Lines 21-22: There is some overlap between these variables and at this point it is unclear how they are uniquely distinguished. For example, incoming solar radiation will vary with slope, aspect, and vegetation, all of which are variables listed here. Is there something unique about "solar radiation" that you should list it here? Does it vary with atmospheric conditions? Please clarify.

There is clear multi-collinearity between the correlated variables, however we used all of them initially in the model to select which were the most powerful predictive variables. Solar radiation was initially included because mid-winter snow melt events are very common in the warm maritime snowpacks and we thought perhaps solar radiation (a strong driver of snow ablation) would be more significant than slope and aspect alone. Although we have only included the two most dominant explanatory variables in this final analysis to prevent overfitting the BRT model.

- Page 5, Line 16: Presumably the model was run at a sub-daily time step (necessary for physical models), but the model provided outputs on a daily basis. Please rephrase.

The SnowModel input data were developed from model runs at a daily time step using data which were collected hourly and integrated to the daily time step. The specific methods for these model outputs are described in this paper referenced in the manuscript, Sproles, E. A., Nolin, A. W., Rittger, K., and Painter, T. H.: Climate change impacts on maritime mountain snowpack in the Oregon Cascades, Hydrology and Earth System Sciences, 17, 2581-2597, 10.5194/hess-17-2581-2013, 2013.

- Page 5, Line 24: Please provide more information about how finer resolution spatial data (e.g., 10-m elevation, 30-m land cover data, etc.) were aggregated to 100-m, and how coarser resolution spatial data (e.g., the 250-m NDVI data) were resampled/downscaled to 100-m.

The following information was included, "All spatial data were converted to the same projection and spatial resolution: NAD83, UTM Zone 10, and a 100-m grid cell size using bilinear interpolation for continuous data and nearest neighbor interpolation for discrete data. Spatial data were processed using ArcGIS 10.1."

- Page 5, Line 24: You already cited the maker/city of ArcGIS, so I am unsure if you need to do it again. Yes thank you, this change was made.

- Page 5, Line 27: Did you use the publically available locations of the SNOTEL sites? The publically available coordinates are imprecise.

No, we used locations obtained from the Oregon NRCS resource managers.

- Page 6, Lines 2-4: Again, I question the independence of the physiographic predictor variables.

We have addressed the multi-collinearity of the physiographic predictor variables by only using the most powerful explanatory variables and excluding all other potentially correlated variables with weak predictive capacity even if they were considered significant in the original model.

- Page 6, Line 21: Add "a" before "set".

This change was made, thank you for catching this error.

- Page 6, Line 23: Revise to say "and public lands where the presence. . .".

This change was made.

- Page 6, Line 27: Did you test for normality? Perhaps include the skew and kurtosis. There is a bit of a skew toward higher SWE volume at the higher elevations, which is why I ask.

We have included the skewness and kurtosis values for the SWE distribution across the elevational gradient in the McKenzie River Basin.

- Page 7, Lines 1-2: Consider including a separate SWE volume line in Figure 3 for the climate change scenario. This will provide another way of showing the shift toward higher elevations above the SNOTEL sites (in addition to the spatial plots in Figure 2).

We have excluded this analysis for this revised manuscript.

- Page 7, Line 3: Is this SWE range measured or modeled at the SNOTEL sites? Please state.

This analysis and associated results were removed from this revised manuscript.

- Page 7, Line 11: Please include units on the RMSE.

RMSE units now included.

- Page 7, Line 12: How much variance did elevation explain? Please quantify.

The following statement was included in the revised manuscript, "Elevation explained the most variance in modeled SWE across the basin, and is the primary driver of all snow classes (2009 BRT model with only elevation; R2 = 0.91, p-value < 0.01)."

- Page 7, Line 15: Recommend using a different word than "believed". Also, it is possible to test the influence of the Three Sisters – just exclude those points in the BRT anaylsis and compare the resulting regression trees.

Latitude explains less than 1% of the variability in the final BRT model and therefore was removed to prevent overfitting of the final model.

- Page 7, Line 20: Should this be 6%? 1.05/0.99 = 1.061 or 6.1%.

These statistics are now changed because we used input data from the actual date of peak SWE instead of using assuming peak SWE was 01 April.

- Page 7, Lines 24-25: Check the sentence: "Although these areas. . .. Above 1791 m." This does not appear to be a complete sentence.

The sentence was changed to the following, "Deep snowpack at the highest elevations only cover a small aerial extent of the MRB, which resulted in decreasing contribution of total basin-wide SWE above approximately 1700 m during the average and above average snow years. In contrast, during the low snow year, the highest elevation classes contributed the most to total basin-wide SWE (Figure 5)."

- Page 8, Line 1: Please clarify which model when you state "greatest error in the model". I think it is the BRT model. Also, the use of the term "error" implies that the SnowModel output is "truth" in the comparison, which may be tenuous. Consider using some language like "difference between models" in this context.

We included three years of input data in this revised manuscript and compared the BRT models using omission and commission statistics to compute overall accuracy between years. The language has been changed throughout the manuscript to evaluate "differences between models". We included the following statement, "The final optimal BRT model from the normal snow year (2009) applied to the high snow year (2008) demonstrated an overall accuracy of 63%, whereas the BRT model applied to the low snow year (2005) demonstrated an overall accuracy of 26% (Table S1, S2). The BRT model performed well across the low and high elevations, where errors of omission and commission were generally lowest (Table S1, S2). Although across the mid-elevations which consist of a patchwork of forest harvest and fire disturbance, were

the areas with the greatest error between the BRT models. . ."

- Page 8, Lines 22-28: This is more appropriate for the discussion section, not the results section.

This was moved to the discussion section which now includes the following, "The ForEST network contributes to the existing SNOTEL network to explicitly investigate snow-vegetation-climate interactions across the range of elevations and forest types in the watershed. The ForEST network is unique in that the monitoring site locations were selected based on statistical classification and geospatial analysis, rather than subjective methods that may incorporate bias. The paired forest-open land cover site selection process alone is not unusual, and has already led to important understanding of key sub-canopy snow processes (Storck et al., 2002; Golding and Swanson, 1986), but here, it has been further validated using coupled physically-based spatially-distributed snow model input data and non-parametric BRT statistical modeling. After five consecutive years of snow monitoring, we have created a valuable and detailed dataset of snow accumulation, snow ablation, and snowpack energy balance that spans the spatial variability in forest and open land cover types across an elevational gradient. The inter-annual consistency in patterns of snow surface energy budget and snow-vegetation interactions across the elevational gradient of the ForEST network suggest that the data are representative of key snow accumulation processes in the MRB (Figure 6). "

- Page 9, Line 12: Add "a" before "key role".

This change was made.

- Page 9, Line 20: Improper semi-colon usage. You can safely remove it, or break the sentence into two here.

This change was made.

- Page 9, Line 23: Replace "does incorporate" with "incorporates".

This change was made.

- Page 9, Lines 23-26: This is a long and overly complicated sentence. Please rephrase and/or revise into shorter sentences.

The following sentence has been included instead," The paired forest-open land cover site selection process has already led to important understanding of key sub-canopy snow processes (Storck et al., 2002; Golding and Swanson, 1986). But here, it has been further validated using coupled physically-based spatially-distributed snow model input data and non-parametric BRT statistical modelling across a forested montane watershed."

- Page 10, Line 19: If a hypothesis is validated, is it still a "working hypothesis"? The word choice is puzzling here.

The sentence has been changed to the following, "However in the rugged and densely forested mountain regions of the western Cascade Mountains where there are few alternatives to modeling spatially distributed SWE, this approach provides a validated hypothesis to guide representative and objective snow monitoring efforts."

TABLE AND FIGURE COMMENTS

- Figure 2 caption: Replace "in shown" with "is shown".

This change was made.

- Figure 2 caption: Please define the units of SWE.

This change was made.

- Figure 2: If April 2009 is an average year (page 5, line 19) and the climate change scenario is a 2 degree C perturbation to an average year, why is the maximum SWE lower in April 2009 (4.31) than in the climate change scenario (5.03)?

This analysis was omitted from this revised manuscript.

[Figure]

- Table 1: What is the logic of the organization of snow classes in Table 1? It generally goes from low to high elevation, except the 977 to 1199 elevations are not in order. Please rectify.

Table 1 has been simplified by elevation and includes all three years of BRT-derived snow classes.

- Table 1: Should snow class 1 read "977-1199" instead of "977-199"?

This change was made.

- Table 1: Consider showing statistics with each snow class to record how well the regression works in that group.

Table 1 is already very busy, and would we prefer to not include additional information that may make the table more difficult to interpret.

- Table 1: What is the purpose of having a binary vegetation class (forest vs. open) and forest canopy cover (CC) predictor variables? Would it not be more straightforward to just include CC and let the BRT tell us when/where the binary distinction dominates the SWE response?

To define a parsimonious and interpretable final BRT model we have only included elevation and land cover in the final BRT model. Land cover type was only slightly more predictive than NDVI in the BRT model, which may be because BRT models tend to optimize categorical variables. Also the slight variability between the BRT classes in the ranges of a continuous variable like NDVI or % canopy cover is confusing when comparing between years. Therefore we decided to only include the binary land cover data in the final BRT model that is clearly defined across all snow classes.

- Table 1: In some (but not all) cases, there is an overlap in the elevation. Is a location at 1426 m elevation in the open in snow class 11 or snow class 13?

The previous snow classes have been slightly redefined because we are using data

from the actual date of peak SWE instead of 01 April. There is no overlap in the elevation between classes.

- Figure 3: Please use a superscript for cubic km on the left y-axis.

This change was made.

- Table 2: It is unconventional to have negative standard deviation or coefficient of variation. Please make these positive. Also, are the CV numbers correct? They should be the SD/Mean, but that does not appear to be the case here.

Table 2 was omitted from the revised manuscript. Instead of comparing CV numbers we are using omission vs commission statistics to evaluate the accuracy of spatial variability of SWE between years.

The results of this accuracy assessment are discussed in the text, and included in the supplementary tables 1 and 2.

- Table 2 caption: Please include the units of SWE differences here.

Table 2 was omitted from the revised manuscript.

Thank you very much for your consider review of our manuscript.

Sincerely, Kelly Gleason

Please also note the supplement to this comment:
http://www.hydrol-earth-syst-sci-discuss.net/hess-2016-317/hess-2016-317-AC3-supplement.pdf
* * *
[Figure]

Table S1. Accuracy assessment matrix comparing the BRT classes derived from the normal snow year 2009 with those from the high snow year 2008. Overall there is less error in the lowest and highest elevation BRT classes, whereas the mid- elevations there is more error between models. Many classes were reassigned when the BRT model was rerun between years, underestimating the accuracy of the overall spatial variability between models.

| BRT Class 2008 \ 2009 | 1 | 2 | 3 | 4 | 5 | 6 | 7 | 8 | 9 | 10 | 11 | 12 | 13 | 14 | 15 | 16 | 17 | 18 | 19 | 20 | 21 | Comission error (%) |
|---|---|---|---|---|---|---|---|---|---|---|---|---|---|---|---|---|---|---|---|---|---|---|
| 1 | 55402 | 6035 | | | | | | | | | | | | | | | | | | | | 10 |
| 2 | | 16467 | | | | | | | | | | | | | | | | | | | | 0 |
| 3 | | 369 | 22960 | | | | | | | | | | | | | | | | | | | 2 |
| 4 | | 52 | | 3930 | | | | | | | | | | | | | | | | | | 1 |
| 5 | | | | | 9879 | | | | | | | | | | | | | | | | | 0 |
| 6 | | | | | 5486 | | | | | | | | | | | | | | | | | 100 |
| 7 | | | | | | 3232 | 3232 | | | | | | | | | | | | | | | 50 |
| 8 | | | | | | | | 4667 | | | | | | | | | | | | | | 0 |
| 9 | | | | | | | | | 2524 | | | | | | | | | | | | | 0 |
| 10 | | | | | | | | 2053 | | 4007 | | | | | | | | | | | | 34 |
| 11 | | | | | | | | | | 5276 | 5740 | | | | | | | | | | | 48 |
| 12 | | | | | | | | | 486 | | | 2900 | | | | | | | | | | 14 |
| 13 | | | | | | | | | | | 1965 | 339 | 5421 | | | | | | | | | 30 |
| 14 | | | | | | | | | | | | | 5252 | 4338 | 617 | | | | | | | 57 |
| 15 | | | | | | | | | | | | | | 13692 | 1948 | 719 | | | | | | 88 |
| 16 | | | | | | | | | | | | | | | | 10260 | 14155 | | | | | 58 |
| 17 | | | | | | | | | | | | | | | | | 23580 | | | | | 100 |
| 18 | | | | | | | | | | | | | | | | | | 5931 | 705 | | | 100 |
| 19 | | | | | | | | | | | | | | | | | | | 1850 | | | 100 |
| 20 | | | | | | | | | | | | | | | | | | | 1057 | 1025 | | 51 |
| 21 | | | | | | | | | | | | | | | | | | | | 2039 | | 0 |
| Omission error (%) | 0 | 28 | 0 | 0 | 36 | 100 | 0 | 31 | 16 | 57 | 26 | 10 | 49 | 76 | 24 | 7 | 100 | 100 | 100 | 71 | 33 | Overall accuracy 63 |

**Fig. 1.** Supplemental Table 1_Accuracy Assesment

[Figure]

Table S2. Accuracy assessment matrix comparing the BRT classes derived from the normal snow year 2005 with those from the high snow year 2008. Overall there is less error in the lowest and highest elevation BRT classes, whereas the mid- elevations there is more error between models. Many classes were reassigned when the BRT model was rerun between years, underestimating the accuracy of the overall spatial variability between models.

| BRT Class 2009 / 2005 | 1 | 2 | 3 | 4 | 5 | 6 | 7 | 8 | 9 | 10 | 11 | 12 | 13 | 14 | 15 | 16 | 17 | 18 | 19 | 20 | 21 | Comission error (%) |
|---|---|---|---|---|---|---|---|---|---|---|---|---|---|---|---|---|---|---|---|---|---|---|
| 1 | 55402 | 22923 | 22960 | 3930 | 15365 | 3232 | 6013 | 3365 | 2243 | | | | | | | | | | | | | 59 |
| 2 | | | | | | | | 3355 | | 9283 | 5840 | | | | | | | | | | | 100 |
| 3 | | | | | | | 767 | | | | | 2900 | | | | | | | | | | 100 |
| 4 | | | | | | | | | | 1965 | | | 9212 | 12939 | | | | | | | | 100 |
| 5 | | | | | | | | | | | | | | 5091 | 757 | 3973 | | | | | | 100 |
| 6 | | | | | | | | | | 339 | 1461 | | | 1808 | 879 | | | | | | | 100 |
| 7 | | | | | | | | | | | | | | | | 3718 | | | | | | 100 |
| 8 | | | | | | | | | | | | | | | | | 2194 | | | | | 100 |
| 9 | | | | | | | | | | | | | | | | | 3622 | | | | | 100 |
| 10 | | | | | | | | | | | | | | | | | 2697 | | | | | 100 |
| 11 | | | | | | | | | | | | | | | | | 3702 | | | | | 100 |
| 12 | | | | | | | | | | | | | | | | | 1815 | | | | | 100 |
| 13 | | | | | | | | | | | | | | | | | | 7239 | | | | 100 |
| 14 | | | | | | | | | | | | | | | | | | 4776 | | | | 100 |
| 15 | | | | | | | | | | | | | | | | | | 4045 | | | | 100 |
| 16 | | | | | | | | | | | | | | | | | | 2347 | | | | 100 |
| 17 | | | | | | | | | | | | | | | | | | 3253 | | | | 100 |
| 18 | | | | | | | | | | | | | | | | | | 1923 | 512 | | | 21 |
| 19 | | | | | | | | | | | | | | | | | | | 3857 | | | 0 |
| 20 | | | | | | | | | | | | | | | | | | | 1562 | 3612 | 421 | 35 |
| 21 | | | | | | | | | | | | | | | | | | | | | 2643 | 0 |
| Omission error (%) | 0 | 100 | 100 | 100 | 100 | 100 | 100 | 100 | 100 | 100 | 100 | 100 | 100 | 100 | 100 | 100 | 100 | 92 | 35 | 0 | 14 | |
| | | | | | | | | | | | | | | | | | | Overall accuracy | | | | 28 |

**Fig. 2.** Supplemental Table 2_Accuracy Assesment

**Supplement:**

[revised manuscript text omitted]

---

## Author Response (AR2)

**Authors' Response**

**Developing a representative snow monitoring network in a forested mountain watershed**
**Kelly E. Gleason, Anne W. Nolin, and Travis R. Roth**

*Dear Editor and Referees,*

*Thank you very much for your considerate review of our research paper entitled, "Developing a representative snow monitoring network in a forested mountain watershed", for submission in Hydrology and Earth System Sciences. Your comments throughout the first and second iterations of this manuscript have greatly improved the quality of the final manuscript. Please see our responses to each of your comments below, as well as in the final manuscript.*

*Sincerely,*
*Kelly*

**Point-by-Point Response to Reviews**

**Comments by Anonymous Referee #1**

I would like to thank the authors for making substantive revisions to their original manuscript. It is a much more focused paper as a result, which was very necessary after my original appraisal. In particular, omitting the SNOTEL climate change analysis is a sensible refinement, as well as focusing on peak SWE rather than a rule of thumb, but somewhat arbitrary, 1 April SWE value. Additionally using three years of data, rather than just 2009 is a significant improvement. To summarize, I am happy with all the responses made to my original review. However, there are two comments that came out of the response letter which I would like to see included. They are minor, but I think will help future readers of this paper:

1. Could brief information be given on the precipitation partitioning used by Sproles et al. (2013)? While I understand the authors desire for brevity in this section, so they can just get on with using these data, brief knowledge of the changes made to precipitation would provide critical added value to those who only read this paper.

*The information in the methods section has been expanded to include the following statement, "The model was modified by Sproles et al., (2013) to account for rain/snow precipitation phase partitioning using a linear function of air temperature from -2° to 2° C (USACE, 1965), and snow*

*albedo decay in forested landscapes using an empirically-based exponential decay function (Burles and Boon, 2011)."*

2. Two comments made in the response letter were very helpful to my understanding of the scope of this paper: "We don't expect the BRT model to predict actual SWE volume on the landscape, but to predict the spatial distribution of similar SWE characteristics across the landscape, and we believe we have achieved this goal in this revised manuscript." and "We did not expect to advance scientific knowledge, but to provide an objective technique for distributing point based monitoring locations which represent the spatial variability across the watershed." Neither statement made its way as explicitly into the paper as it was made in the response letter. To focus the reader on the scope of this work, I would like to see the main thrust of both of these statements added to the final paragraph of the introduction. That way the limitations to the goals of this study (not predicting SWE volume, but instead spatial characteristics of SWE; the novelty is through application of a method rather than any revolutionary methodological developments) would be clear.

*We appreciate the need for clarity related to two major points requested by Referee #1, including, 1) that this method attempts to capture the spatial characteristics of SWE, and 2) that the novelty of this research is in the application of the method. We explain in the last paragraph of the introduction that this method aims to capture the spatial variability of SWE relative to landscape physiography, in the following statements, "To objectively identify optimal site locations to distribute a snow monitoring network which explicitly captures the spatial variability of snow accumulation relative to the physiographic landscape we used a combination of physically-based, statistical, and geospatial models. This paper presents this objective and relatively simple methodology to distribute a snow monitoring network which captures landscape driven spatial variability in snow accumulation and includes four major objectives:"*

*Also, we explain the novelty of this research in the Conclusions section of the manuscript which includes the following statement, "The novelty of this research stems from the application of the method, where by the coupling of a traditional BRT classification process with a validated physically-based spatially distributed model, we improved snow observational network design in a forested montane watershed." Particularly because of the comments by Referee #3 that, "this study has novel and useful contributions", we believe this explanation is best in the Conclusions section of the manuscript.*

**Comments by Anonymous Referee #3**

SUMMARY

The authors did an excellent job in revising the manuscript to address my concerns from the first review. The reframing of the analysis to focus on three different snow years (rather than climate change projections) alleviated the concerns I raised about the comparability of different years on a fixed date. Likewise, they adequately addressed my concerns about multi-collinearity. Contrary to other reviewers, I find this study has novel and useful contributions. I recommend publication after considering a few additional minor comments (below).

TECHNICAL COMMENTS

- P1 L12: Should read "an average snow year".

*The suggested change has been made in the abstract.*

TABLE AND FIGURE COMMENTS

- Figure 1: Either include the Columbia basin located in Canada in the shaded region, or note in the caption that the shaded region only includes the Columbia basin in the USA. Currently it suggests the Canada-USA border is coincident with the basin boundary, when it is not.

*The entire Columbia river basin (including the Canadian section) is now included in Figure 1.*

- Figures 2-4: Consider arranging the three years more consistently across figures. For example, Figure 3 has 2008, 2009, then 2005 while the others have 2009, 2008, then 2005. This is a minor suggestion.

*In Figures 2-4, the study years are now arranged 2009, 2008, then 2005 for consistency.*

- Figure 2: The legends are too small and make it hard to read without zooming. Please enlarge.

*The legends in Figure 2 and Figure 4 have been enlarged.*

- Figure 3: It is not clear in the figure or caption which vertical axis corresponds to the gray dashed line, the elevation fraction (or percent?). Otherwise, this is an outstanding figure.

*This has been clarified in the figure caption which includes the following text, "The dashed grey line indicates the % / 100 of the area represented by each 100 m elevation band, and its values are associated with the left y-axis."*

- Figure 5: Consider reducing the size of the star symbols a little bit.

*The size of the star symbols has been reduced by 30%.*

**List of All Relevant Changes**

1. *The information in the methods section was expanded to further explain the precipitation phase partitioning used in the modelled data.*
2. *The aim of this research to capture the spatial variability in SWE, and the novelty of this research in the application of the method were clarified in the introduction and conclusions sections of the manuscript.*
3. *The typo in the abstract was corrected.*
4. *The entire Columbia River Basin was included in Figure 1.*
5. *The study years have been arranged consistently in all figures from 2009, 2008, and 2005*
6. *The legends in Figures 2 and 4 were enlarged.*
7. *The caption Figure 3 was corrected to include the association of the dashed grey line with the left y-axis.*
8. *The stars in Figure 5 have been enlarged.*

[revised manuscript text omitted]